# Restricting datasets to classifiable samples augments discovery of immune disease biomarkers

Gunther Glehr [1], Paloma Riquelme [1], Katharina Kronenberg [1], Robert Lohmayer[2], Víctor J. López-Madrona [3], Michael Kapinsky[4], Hans J. Schlitt [1], Edward K. Geissler[1], Rainer Spang[5], Sebastian Haferkamp[6] & James A. Hutchinson [1]✉

Immunological diseases are typically heterogeneous in clinical presentation, severity and response to therapy. Biomarkers of immune diseases often reflect this variability, especially compared to their regulated behaviour in health. This leads to a common difficulty that frustrates biomarker discovery and interpretation – namely, unequal dispersion of immune disease biomarker expression between patient classes necessarily limits a biomarker's informative range. To solve this problem, we introduce dataset restriction, a procedure that splits datasets into classifiable and unclassifiable samples. Applied to synthetic flow cytometry data, restriction identifies biomarkers that are otherwise disregarded. In advanced melanoma, restriction finds biomarkers of immune-related adverse event risk after immunotherapy and enables us to build multivariate models that accurately predict immunotherapy-related hepatitis. Hence, dataset restriction augments discovery of immune disease biomarkers, increases predictive certainty for classifiable samples and improves multivariate models incorporating biomarkers with a limited informative range. This principle can be directly extended to any classification task.

The immune system detects pathological challenges with exquisite sensitivity and specificity, enabling it to mount appropriate protective responses[1]. Widely distributed immune cell subsets are responsible for sensing pathogens, tissue injury and cellular stress through diverse receptor systems[2–4]. These disease-related signals are then amplified through humoral and cellular cascades that stimulate migration, expansion and activation of particular effector cell populations[5]. By capturing information about the precise nature of immune responses, we can draw inferences about the triggering event, allowing us to develop diagnostic or prognostic models to guide personalised treatment decisions[6].

Flow cytometry is a sophisticated, fast and relatively inexpensive method for analysing the properties of single cells from a cell suspension[7]. In clinical practice, flow cytometry is commonly used to profile leukocyte subset distribution in patient blood samples, especially in the context of haematological malignancies and infectious diseases[8]. Modern cytometers simultaneously collect data about expression of multiple proteins in single cells, while also allowing us to interrogate many millions of cells from a single sample[9]. This enables accurate identification of narrowly defined cell subsets, including rare populations, as well as broadly surveying many leukocyte subsets[10]. This rich information is captured as a data matrix for each sample with

[1]Department of Surgery, University Hospital Regensburg, Regensburg, Germany. [2]Algorithmic Bioinformatics Research Group, Leibniz Institute for Immunotherapy, Regensburg, Germany. [3]Aix Marseille Univ, INSERM, INS, Inst Neurosci Syst, Marseille, France. [4]Beckman Coulter Life Sciences GmbH, Krefeld, Germany. [5]Department of Statistical Bioinformatics, University of Regensburg, Regensburg, Germany. [6]Department of Dermatology, University Hospital Regensburg, Regensburg, Germany. ✉e-mail: james.hutchinson@klinik.uni-regensburg.de

an unordered number of rows corresponding to cells and a defined number of cell antigen expressions as columns[11].

Applications of flow cytometry in clinical diagnostics are growing rapidly[12]. Of special interest, recent reports claim that immunophenotyping of peripheral blood leukocytes can be used to predict immune-related adverse events (irAE) following immune checkpoint inhibitor (ICI)-therapy[13–16]. Combined treatment with anti-PD-1 (Nivolumab) and anti-CTLA-4 antibody (Ipilimumab) is now first-line therapy for many patients with unresectable metastatic melanoma[17]. Its effectiveness is remarkable in terms of clinical response rates, progression-free survival and overall survival; however, immune-mediated complications, such as colitis or hepatitis, present a significant clinical concern[18]. Life-threatening reactions are uncommon[19], but irAE often require interruption or discontinuation of immunotherapy, and introduction of glucocorticoids or non-steroidal immunosuppressants[20]. Clinically applicable, robust biomarkers to guide irAE prevention or treatment strategies in patients would be useful[21].

Extracting reliable predictive information from flow cytometry measurements is difficult because disease-related changes are often small compared to typical biological and technical variations[22]. This is especially true when investigating systemic changes in peripheral blood samples for signals that reflect localised disease[23]. Consequently, we often rely upon computational methods to perceive small and multivariate, but consistent changes between patient samples[24]. Most current approaches entail identifying cell populations with clustering methods like FlowSOM[25], extracting sample-wise cell frequencies from each cluster and then comparing between samples to identify significantly differentially represented cell subsets[26]. Alternatively, some methods identify disease-related changes at a single-cell level[27].

Compared to the tightly regulated homeostasis of health, immunological diseases are inherently more variable[28]. Generally speaking, it follows that immune disease-related biomarkers are more variably expressed in disease than health[29,30]. As we show, this fundamental biological insight is important because overlapping biomarker expression with unequal dispersion between patient classes necessarily implies a range of biomarker values with no discriminatory potential. This problem is exaggerated when biomarker distributions with unequal dispersion substantially overlap between two patient classes, such as health and disease. Critically, we often find that disease-related differences in immunological biomarkers are small in relative and absolute terms[31]. This inconvenient and unintuitive property, which is typical of flow cytometry data, masks informative biomarkers in discovery studies and limits their clinical utility[32].

In this report, we examine the problem of finding and interpreting disease biomarkers with a restricted range of informative values from an immunologist's perspective. To do this, we must first disambiguate some key terms. Properties of single cells measured by flow cytometry, such as cell lineage-associated surface antigen expression, will be called "cell antigens". We reserve "biomarker" to mean a sample-related quantity, such as cell subset frequency, that is relevant to sample classification, hence diagnoses. The distribution of biomarker values within a set of patient samples is described by its probability density function, or simply "density". Throughout this article, we present plots of densities that compare biomarker expression in patient subgroups: these should not be mistaken for histograms showing antigen expression within samples.

We provide a computational method to optimally restrict biomarkers to their informative range, which makes them easier to discover and interpret. The power of dataset restriction is demonstrated through its application to flow cytometry results from patients with metastatic melanoma receiving Ipilimumab plus Nivolumab (Ipi-Nivo) therapy. For each biomarker, we calculate a restricted standardised AUC (rzAUC) for every measured value by splitting the sample set into

biomarker^HIGH and biomarker^LOW parts. We define the optimal restriction according to the maximum absolute rzAUC of either the biomarker^HIGH or biomarker^LOW part. We then assign a permutation p-value to the optimal rzAUC. Finally, we leverage the adapted range of all restricted biomarkers in a multivariate (random forest) model by forcing decision tree cuts within each informative range.

In essence, restriction identifies the informative range of a biomarker, allowing us to segregate datasets into classifiable and unclassifiable samples. Importantly, using information about the informative range of biomarkers typically leads to superior multivariate models. We qualify our method using realistically simulated flow cytometry data, then apply it to real T cell subset analyses to discover biomarkers of irAE risk in patients receiving immunotherapy for advanced melanoma. Using a restricted dataset, we were able to train and prospectively validate a multivariate model to predict immunotherapy-related hepatitis, which failed when using unrestricted data. Our computational methods can be directly applied to other types of data, not limited to transcriptomic, proteomic, mass cytometric, and microbiomic information.

## Results

### Two-class distributions resulting in skewed ROC curves

We begin by showing how the distribution of a discriminatory biomarker that differs in expression between diseased (patients) and unaffected (controls) individuals results in skewed receiver operating characteristic (ROC) curves. ROC curves relate the true positive rate (TPR) and false positive rate (FPR) for a disease biomarker at every data point in a two-class classification problem. The area under a ROC curve (AUC) is often used as a measure of the discriminatory capacity of a disease biomarker[33]. Throughout this report, we illustrate distributions of biomarker expression within classes by plotting probability densities. Densities are normalised to 1 within each class, so the appearance of these plots is independent of class size (see Supplementary Note 1). In the following sections, we consider hypothetical biomarkers whose expression is normally distributed $\mathcal{N}(\mu,\sigma^2)$ with mean $\mu$ and variance $\sigma^2$.

Perfectly discriminatory biomarkers result in concave ROC curves with an AUC = 1 (Fig. 1a). For imperfect biomarkers, where there is overlap between the distributions of a disease biomarker expression in patient and control populations, provided that variance is equal in both classes, the ROC curve is symmetric about the anti-diagonal with $1 > AUC > 0.5$. In the hypothetical example, biomarker expression is normally distributed with equal variances in the patient $\mathcal{N}(6,1)$ and control $\mathcal{N}(5,1)$ populations, but the mean expression is higher in patients (Fig. 1b). Entirely uninformative biomarkers result in straight diagonal ROC curves with an AUC = 0.5 (Fig. 1c).

Interpreting the area under a ROC curve is more complicated when comparing overlapping biomarker distributions with unequal variances that result in ROC curves skewed around the anti-diagonal. Our first hypothetical example of a skewed ROC curve shows that normally distributed, overlapping biomarker distributions with a higher mean and variance in the patient population compared to controls lead to a right-skewed ROC curve that crosses the diagonal in a region corresponding to low biomarker expression values (Fig. 1d). It is generally true that normally distributed populations with different variances result in ROC curves that cross the diagonal[34]. To illustrate this point, we simulated 200 samples by drawing random values from Normal distributions to show how varying the mean and variance of biomarker expression in patient and control distributions affects the shape and AUC of ROC curves (Supplementary Movies 1 and 2). In the context of clinical diagnostics, biomarkers of immune diseases usually reflect a change between tightly regulated homeostasis in health and a disturbed, higher-variability condition in disease. Coupled with the fact that disease-associated changes in cell subset frequencies in the blood are typically small, it is perhaps unsurprising that disease

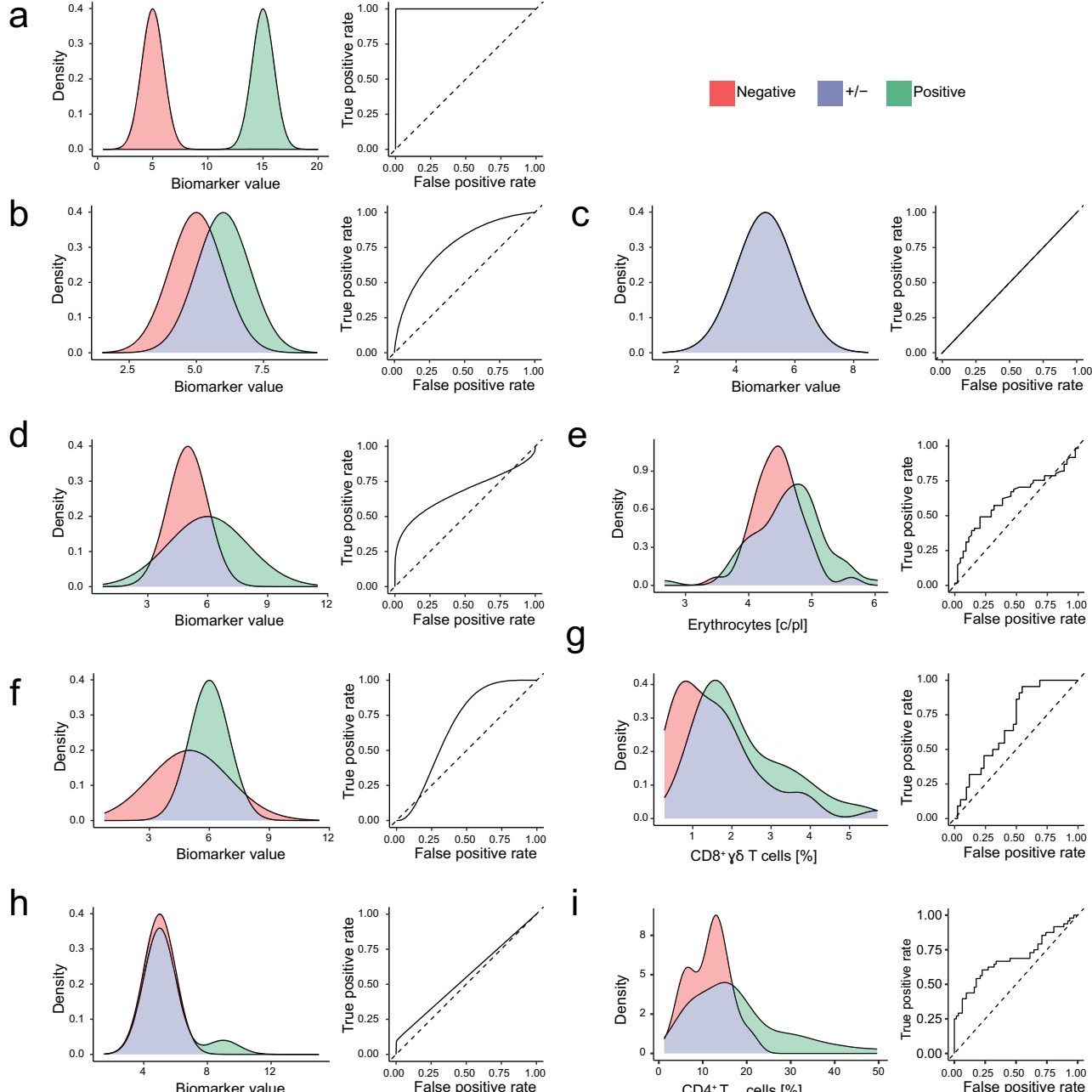

**Fig. 1 | Two-class distributions resulting in asymmetric ROC curves.** We present hypothetical and real-world examples of biomarker distributions in two classes that represent sets of patients with different clinical outcomes. The distribution of values from the positive (i.e. diseased) class is coloured green, and the negative (i.e. control) class is coloured red. Overlapping densities are coloured purple. For each example, we present the corresponding ROC curve. **a** A hypothetical example of a perfectly discriminatory biomarker with negative $\mathcal{N}(5,1)$ and positive $\mathcal{N}(15,1)$ populations that give rise to a symmetrical ROC curve. The area under the ROC curve (AUC) is 1.0. **b** A hypothetical example of substantially overlapping biomarker distributions in the negative $\mathcal{N}(5,1)$ and positive $\mathcal{N}(6,1)$ populations that give rise to a symmetrical ROC curve with AUC = 0.76. **c** A hypothetical example of an uninformative biomarker distribution with negative $\mathcal{N}(5,1)$ and positive $\mathcal{N}(5,1)$ populations that give rise to a diagonal ROC curve with AUC = 0.5. **d** A hypothetical example of substantially overlapping biomarker distributions in the negative $\mathcal{N}(5,1)$ and positive $\mathcal{N}(6,2)$ populations with unequal variance that gives rise to right-skewed ROC curve with AUC = 0.67. **e** A real-world example of substantially overlapping distributions of absolute erythrocyte counts with unequal variance in patients with metastatic melanoma who responded ($n = 61$) or did not respond ($n = 44$) to combined Ipi-Nivo therapy. We observe a right-skewed ROC curve with AUC = 0.62. **f** A hypothetical example of substantially overlapping biomarker distributions in the negative $\mathcal{N}(5,2)$ and positive $\mathcal{N}(6,1)$ populations with unequal variance that gives rise to a left-skewed ROC curve with AUC = 0.67. **g** A real-world example of substantially overlapping distributions of CD8+ γδ T cells with unequal variance in patients with metastatic melanoma who did ($n = 22$) or did not ($n = 42$) develop treatment-related hepatitis after Ipi-Nivo therapy. We observe a left-skewed ROC curve with AUC = 0.69. **h** A hypothetical example of substantially overlapping biomarker distributions in the normally distributed negative $\mathcal{N}(5,1)$ and bimodally distributed positive populations. In this example, the positive population comprises 10% cases with elevated biomarker expression $\mathcal{N}(9,1)$ and 90% cases with unaltered biomarker expression $\mathcal{N}(5,1)$. Heterogeneity in the diseased cases gives rise to a right-skewed ROC curve with AUC = 0.55. **i** A real-world example of a phenotypically heterogeneous set of patients with metastatic melanoma who did ($n = 48$) or did not ($n = 62$) develop treatment-related hepatitis after Ipi-Nivo therapy. A subset of these patients exhibited a baseline expansion of CD4+ $T_{EM}$ cells that was likely driven by subclinical cytomegalovirus (CMV) reactivation. Consequently, CD4+ $T_{EM}$ cell frequency before therapy is a weakly discriminatory biomarker of hepatitis risk that gives rise to a right-skewed ROC curve with AUC = 0.64.

biomarkers measured by flow cytometry frequently result in skewed ROC curves[13]. In support of this assertion, we present a real-world example of a right-skewed ROC curve with a low AUC (Fig. 1e). Specifically, this example shows that erythrocyte counts were elevated in baseline blood samples from patients with metastatic melanoma who responded to Ipi-Nivo therapy compared to non-responders.

Left-skewed ROC curves arise when the negative population has a lower mean, but higher variance than the positive population (Fig. 1f). We find a real-world example in the previously unreported association between CD8$^+$ γδ T cells and hepatitis risk after combined Ipi-Nivo therapy (Fig. 1g and Supplementary Fig. 2). In this case, the higher variance of the control population might be due to technical imprecision in quantifying a rare cell population, since the absolute number of CD8$^+$ γδ T cells in blood was only $25.6 \pm 19.3$ c/μl.

We next considered the case of a phenotypically heterogeneous positive population, which could reflect multiple aetiologies leading to a common clinical presentation, different stages of a disease that culminate in a common presentation or different treatment responses. In such scenarios, we expect a bimodal distribution of a disease biomarker in the positive population that leads to a skewed ROC curve (Fig. 1h). We previously reported the identification of a subset of patients with advanced melanoma who developed hepatitis after Ipi-Nivo therapy, which was reliably predicted by cytomegalovirus (CMV) associated expansion of CD4$^+$ effector memory T cells (T$_{EM}$) cells prior to immunotherapy[35]. In our melanoma dataset, we show that using CD4$^+$ T$_{EM}$ frequencies to predict hepatitis after immune checkpoint inhibitor (ICI) therapy leads to a right-skewed ROC curve (Fig. 1i). We know from previous work that baseline CD4$^+$ T$_{EM}$ expansion is only a useful biomarker of hepatitis risk in CMV-seropositive patients, who constituted just 47.3% of our study cohort; therefore, this is a biologically validated example of alternative immunopathologies contributing to a common pathological presentation that impacts biomarker performance.

These three hypothetical distributions and their real-world counterparts demonstrate an important concept in immune biomarker discovery – namely, that a disease biomarker may be highly informative over a restricted range of measured values, but will consistently misclassify samples with biomarker values outside that range. This principle is not only limited to Gaussian distributions but also applies to other distributions, such as the negative-binomial distribution that is often used to model count data (Supplementary Fig. 3). By extension, using AUC across the entire ROC curve to assess predictive performance leads us to disregard potentially informative biomarkers. Clearly, we need a method of finding such biomarkers and defining their valid ranges.

## Dataset restriction is a method to find disease biomarkers

Disease biomarkers that give rise to skewed ROC curves perform well in a subset of samples, which may belong to either the positive or negative class, but are only informative over a certain range. This leads us to the idea that particular samples may be classifiable or unclassifiable according to any given disease biomarker. Here, we present and implement a method of biomarker discovery that relies upon restricting training datasets to classifiable samples[36] (Box 1). In the given example, we compared the distributions of 2500 positive and 2500 negative simulated samples, in which 20% of positive and 2% of negative samples were drawn from a normal distribution $\mathcal{N}(9,1)$ and all other samples were drawn from $\mathcal{N}(6,1)$ (Fig. 2a). This resulted in a right-skewed ROC curve for the complete dataset (Fig. 2b). We first generated two ROC curves for every possible "restriction" of the dataset – explicitly, one for samples above the restriction (biomarker$^{HIGH}$ samples, orange; Fig. 2c, d–f) and one for samples beneath (biomarker$^{LOW}$ samples, blue; Fig. 2c, g–i). Biomarker$^{HIGH}$ samples correspond to the bottom-left part of the complete ROC curve (Fig. 2d). Considering the densities of only biomarker$^{HIGH}$

samples (Fig. 2e), the restricted ROC curve had a superior "restricted" AUC (rAUC) of 0.692 (Fig. 2f). Biomarker$^{LOW}$ samples correspond to the top-right part of the complete ROC curve (Fig. 2g) Here the densities of the positive and negative classes overlapped substantially (Fig. 2h). Consequently, the restricted ROC curve was close to diagonal (Fig. 2i). Notably, restricted densities are not the same as those in Fig. 2a but are instead re-calculated on either biomarker$^{HIGH}$ or biomarker$^{LOW}$ samples. Supplementary Movie 3 helps to visualise the *rAUC* for varying restrictions of the dataset.

Standardising each rAUC according to sample size gave the restricted standardised AUC (rzAUC). The maximum absolute value of rzAUC defined the optimal restriction value (Fig. 2c). In our example, rzAUC was maximal at FPR = 0.258, which corresponded to an optimal biomarker restriction value of 6.8. Consequently, biomarker$^{HIGH}$ samples should be kept and biomarker$^{LOW}$ samples should be discarded – that is to say, biomarker$^{HIGH}$ samples are classifiable, whereas biomarker$^{LOW}$ samples are unclassifiable. In other situations, the positive class may have higher or lower biomarker values, potentially leading to an AUC < 0.5 and accordingly, a negative rzAUC. In Supplementary Fig 4, we show that regardless of which class is labelled positive or negative, our method identifies the same restriction value. In such cases, biomarker$^{HIGH}$ and biomarker$^{LOW}$ rzAUCs are mirrored, meaning the restriction at the optimal absolute rzAUC remains identical. Critically, regardless of biomarker distribution, because areas under ROC curves are independent of class size, it follows that restriction values are also independent of class size[37].

## Restriction identifies classifiable samples in simulated datasets

To test our computational approach, we next applied it to our four preceding examples from Fig. 1 by simulating 100 samples from each class. In the first example, the negative class $\mathcal{N}(5,1)$ and positive class $\mathcal{N}(6,1)$ gave rise to a symmetrical ROC curve with a maximum rzAUC corresponding to FPR = 1; consequently, the optimally informative dataset contained all samples (Fig. 3a). In the second example, the negative class $\mathcal{N}(5,1)$ and positive class $\mathcal{N}(6,2)$ produced a right-skewed ROC curve because the variances were unequal (Fig. 3b). We see that low biomarker values led to a consistent misclassification, indicated by the ROC curve crossing the diagonal. The maximum rzAUC of 5.8 for biomarker$^{HIGH}$ samples indicated that samples with a biomarker value < 4 must be discarded. In the third example, the negative class $\mathcal{N}(5,2)$ and positive class $\mathcal{N}(6,1)$ produced a left-skewed ROC curve (Fig. 3c). Here, high biomarker values led to consistent misclassification; therefore, the ROC curve deviated below the diagonal. The maximum rzAUC of 5.8 for biomarker$^{LOW}$ samples indicated that samples with a biomarker value > 7 must be discarded. In the fourth example, we compared 100 samples from the negative class $\mathcal{N}(5,1)$ and a bimodal positive class consisting of 90 samples from the same distribution $\mathcal{N}(5,1)$, plus 10 samples from a distribution $\mathcal{N}(9,1)$ with a higher mean (Fig. 3d). The resulting right-skewed ROC curve reflected the fact that our simulated biomarker was only informative for higher sample values. Accordingly, the optimal rzAUC of 2.4 for biomarker$^{HIGH}$ samples restricted our dataset to samples with a biomarker value ≥ 6.2. Hence, we demonstrated that our method is able to optimally restrict cleanly simulated patient populations, such that we retain only classifiable samples.

## Synthesising realistic flow cytometry datasets

Realistic synthetic data can be valuable in machine learning, especially in validating analytical methods, calculating experimental sample sizes or data augmentation. Because no generative model already existed, we developed an algorithm to create synthetic flow cytometry datasets (Box 2), which differ from the preceding simulated examples in several key respects – specifically, they comprise multiple covarying biomarkers, incorporate a realistic level of noise, and were adjusted in biologically meaningful ways (Fig. 4). Our web-based interactive gating

# BOX 1

# Restriction

**Input:** Biomarker values $Y_i$ for $i$ in 1, ..., $N$ samples of diseased (positive, $D_i = 1$) or non-diseased (negative, $D_i = 0$) class.

**Output:** Optimal restriction ($r$), informative range (info. range), restricted AUC (rAUC), restricted standardised AUC (rzAUC) and permutation p-value.

**Algorithm:**

1. Calculate ROC curve with true positive rate $S_D(y) := P[Y \geq y \mid D = 1]$ and false positive rate $S_{\bar{D}}(y) := P[Y \geq y \mid D = 0]$,

$$\mathrm{ROC}(t) = S_D(S_{\bar{D}}^{-1}(t)).$$

2. For every restriction $r$:

   2a. Calculate the partial area under the ROC curve

   i. up to the false positive rate $S_{\bar{D}}(r) \equiv \beta(r)$ with

   $$\mathrm{AUC}_{\mathrm{high}}(\beta(r)) = \int_0^{S_{\bar{D}}(r)} \mathrm{ROC}(t)\, dt,$$

   ii. starting from a true positive rate $S_D(r) \equiv 1 - \alpha(r)$ with

   $$\mathrm{AUC}_{\mathrm{low}}(\alpha(r)) = \int_{S_{\bar{D}}(r)}^1 \mathrm{ROC}(t)\, dt - S_D(r)(1 - S_{\bar{D}}(r)).$$

   2b. Calculate restricted AUCs

   $$\mathrm{rAUC}_{\mathrm{high}}(r) = \mathrm{AUC}_{\mathrm{high}}(S_{\bar{D}}(r)) \cdot \frac{1}{S_{\bar{D}}(r)} \cdot \frac{1}{S_D(r)},$$

   $$\mathrm{rAUC}_{\mathrm{low}}(r) = \mathrm{AUC}_{\mathrm{low}}(1 - S_D(r)) \cdot \frac{1}{1 - S_{\bar{D}}(r)} \cdot \frac{1}{1 - S_D(r)}.$$

2c. Calculate approximate statistic for $X$='high' or $X$='low'

$$\mathrm{rzAUC}_X(r) = \frac{\mathrm{rAUC}_X(r) - 0.5}{\sqrt{\mathrm{var}_{H_0}(\mathrm{rAUC}_X(r))}},$$

with $\mathrm{var}_{H_0}(\mathrm{rAUC}_X(r)) \approx \frac{(m_X + n_X + 1)}{12 m_X n_X}$,

where $m_X$ is the number of positive and $n_X$ is the number of negative samples defined as

$$m_{\mathrm{low}} := |\{Y_i \leq r, D_i = 1\}|, \quad m_{\mathrm{high}} := |\{Y_i > r, D_i = 1\}|,$$
$$n_{\mathrm{low}} := |\{Y_i \leq r, D_i = 0\}|, \quad n_{\mathrm{high}} := |\{Y_i > r, D_i = 0\}|.$$

3. Find the optimal restriction value $r_{\mathrm{opt}}$ and informative part $X_{\mathrm{opt}}$ by

$$(r_{\mathrm{opt}}, X_{\mathrm{opt}}) := \underset{r,X}{\mathrm{argmax}}\, (\{|\mathrm{rzAUC}_X(r)|\}).$$

4. Report

$$\mathrm{rAUC}_{\mathrm{opt}} = \mathrm{rAUC}_{X_{\mathrm{opt}}}(r_{\mathrm{opt}}) \text{ and } \mathrm{rzAUC}_{\mathrm{opt}} = \mathrm{rzAUC}_{X_{\mathrm{opt}}}(r_{\mathrm{opt}}),$$

$$\text{info. range} = \begin{cases} -(\infty, r_{\mathrm{opt}}], & \text{if } X_{\mathrm{opt}} = \text{'low'} \\ (r_{\mathrm{opt}}, \infty), & \text{else.} \end{cases}$$

5. Calculate permutation p-values.

   5a. Repeat steps 1-3 for permuted class labels ($D$=1, $D$=0) $n_{\mathrm{total}}$ times.

   5b. Count $n_{\mathrm{above}} := \#(\mathrm{rzAUC}_{\mathrm{opt,permutation}} \geq \mathrm{rzAUC}_{\mathrm{opt}})$.

   5c. Calculate p-value (see methods, $p \approx \frac{n_{\mathrm{above}} + 1}{n_{\mathrm{total}} + 1}$).

---

tree allows readers to synthesise their own flow cytometry data (Supplementary Note 2).

To validate our restriction method, we needed a way of imitating disease-related differences between groups of samples. In the method described above, any effect that changes the proportion of cells in any gates equates to changing the Dirichlet distribution parameters. In the given example, the originally estimated mean proportions are projected onto the gating tree and corresponding Dirichlet distribution for three examples leafs A, B and L. Here, the mean proportion of CD8+ effector memory T cells re-expressing CD45RA ($T_{\mathrm{EMRA}}$) cells was 7.17% (Fig. 4a). Now, instead of determining the number of cells in each leaf gate according to the originally estimated distribution, we generated synthetic cells from a modified Dirichlet distribution in which the mean proportion of CD8+ $T_{\mathrm{EMRA}}$ cells was arbitrarily changed to 33.23% (Fig. 4b). Using our method, changing the proportion of cells in any gate leads to changes in the proportion of cells in all other gates, which we represent by the different intensities of red shading in the gating trees and the different Dirichlet distribution for the three example leaves A, B and L. Three examples of gating generated with a mean proportion of CD8+ $T_{\mathrm{EMRA}}$ cells = 33.23% are provided (Supplementary Fig. 5).

**Applying restriction to realistic synthesised flow cytometry datasets**

We next applied our restriction method to synthetic flow cytometry datasets that incorporated estimated technical and biological noise typical of real-world measurements. Specifically, we generated synthetic samples that gave rise to biomarker distributions similar to the preceding simulated examples (Fig. 5). Artificial disease associations were introduced by changing the frequency of CD4+ $T_{\mathrm{EM}}$ cells, which had a baseline mean proportion of 7.7% among healthy donors. We subsequently extracted CD4+ $T_{\mathrm{EM}}$ cell frequencies relative to CD3+ T cells from all samples by applying our standard gating strategy and then applied our restriction method. Similar to Fig. 3, we simulated biomarker values from normal distributions. We then generated synthetic flow cytometry datasets by setting the CD4+ $T_{\mathrm{EM}}$ cell Dirichlet parameter to each simulated biomarker value.

In the first example, the negative class $\mathcal{N}(7.7,1)$ and positive class $\mathcal{N}(10.7,1)$ gave rise to a symmetrical ROC curve (Fig. 5a). As expected, the results were much noisier than those shown in Fig. 3; nevertheless, the maximum rzAUC = 8.8 corresponded to FPR = 1, so the optimally informative dataset contained all samples. In the second example, the negative class $\mathcal{N}(7.7,1)$ and positive class $\mathcal{N}(8.7,3)$ gave rise to a right-skewed ROC curve (Fig. 5b). The maximum rzAUC = 3.56 led us to retain biomarker$^{\mathrm{HIGH}}$ samples with $\geq$ 4.57% CD4+ $T_{\mathrm{EM}}$ cells. In the third example, the negative class $\mathcal{N}(7.7,3)$ and positive class $\mathcal{N}(8.7,1)$ gave rise to a left-skewed ROC curve (Fig. 5c). The maximum rzAUC = 4.11 led to a restriction of the dataset to biomarker$^{\mathrm{LOW}}$ samples with <6.89% CD4+ $T_{\mathrm{EM}}$. In the fourth example, we compared the negative class $\mathcal{N}(7.7,1)$ and a bimodal positive class comprising 80 samples showing no effect $\mathcal{N}(7.7,1)$ plus 20 samples from a distribution $\mathcal{N}(16.7,1)$ with a higher mean (Fig. 5d). The resulting right-skewed ROC curve with a maximum rzAUC = 4.05 led us to keep biomarker$^{\mathrm{HIGH}}$ samples with $\geq$ 8.12% CD4+ $T_{\mathrm{EM}}$. Hence, our method can appropriately restrict

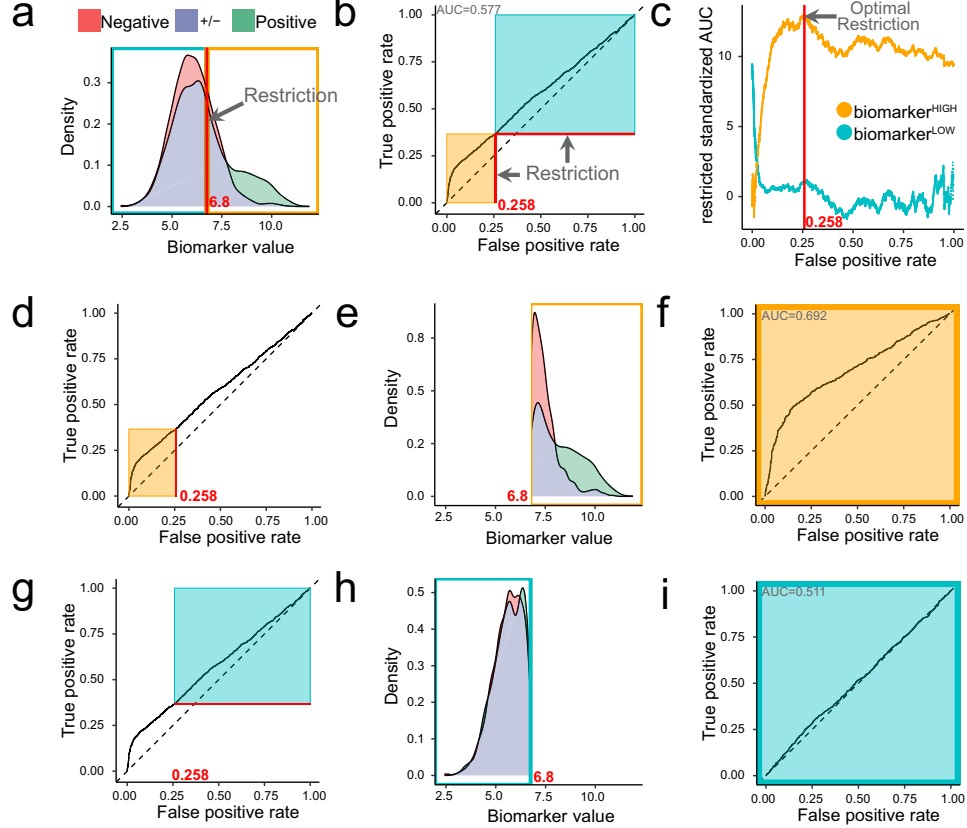

**Fig. 2 | Method to optimally restrict datasets to classifiable samples.** We present a simulated example of biomarker distributions in two classes that represent sets of patients with different clinical outcomes. **a** The distribution of values from the positive ($n = 2500$) class are coloured green and values from the negative ($n = 2500$) class are coloured red; the overlapping density areas are coloured purple. In this example, 20% of positive samples and 2% of negative samples were drawn from a population with elevated biomarker expression $\mathcal{N}(9,1)$. All other samples were drawn from a population with unaltered biomarker expression $\mathcal{N}(6,1)$. The optimal restriction of this dataset lies at a biomarker value of 6.8, which is marked with a red line. Restriction of the dataset defines two subsets of samples: biomarker[HIGH] (orange) and biomarker[LOW] (blue) samples. **b** A complete ROC curve marked at the optimal restriction point (red lines) corresponding to FPR = 0.258. Restricting the ROC curve corresponding to biomarker[HIGH] or biomarker[LOW] samples gives us restricted ROC curves for which restricted AUCs (rAUCs) are calculated. **c** Adjusting the rAUC for the number of samples delimited by the restriction gives the restricted standardised AUC (rzAUC) that can be plotted for biomarker[HIGH] and biomarker[LOW] samples at all possible restriction values. The optimal restriction value is defined as the maximum absolute rzAUC for either the biomarker[HIGH] or biomarker[LOW] samples. **d** A complete ROC curve to illustrate the delimitation of biomarker[HIGH] values (orange rectangle) according to the optimal restriction. **e** Densities of the negative and positive classes after restriction to biomarker[HIGH] values. **f** ROC curve constructed from biomarker[HIGH] samples. **g** A complete ROC curve to illustrate the delimitation of biomarker[LOW] values (blue rectangle) according to the optimal restriction. **h** Densities of the negative and positive classes after restriction to biomarker[LOW] values. **i** ROC curve constructed from biomarker[LOW] samples.

realistically synthesised flow cytometry datasets for symmetric or skewed ROC curves, such that we retain only classifiable samples.

## Restriction method improves findability in realistic synthesised datasets

As explained above, introducing an artificial disease association into realistically synthesised flow cytometry datasets by adjusting the frequency of one cell population (in this case, CD4[+] T[EM] cells) leads to changes in all other nodes in our gating tree. We next asked whether our restriction method could also improve the discoverability of these covariant biomarkers in the synthesised datasets presented above. To do this, we assigned significance values to the rzAUC. The AUC is equivalent to the Mann-Whitney U-statistic[33] and we can extend this equivalence to the rzAUC; however, this does not help assign significance values because optimising for the highest rzAUC introduces a bias (Supplementary Fig 6). Instead, we must calculate permutation p-values[38], which are uniformly distributed as expected after random permutation of labels (Supplementary Fig 7). For each of our four realistic synthesised examples, we calculated permutation p-values using the unrestricted sample set and the optimally restricted sample set for every gated cell population. Figure 6 shows these p-values as scatter plots in which the green-shading demarcates unrestricted p-values > 0.05 and optimally restricted p-values < 0.05 – that is, biomarkers identified as significant using our restriction method, but missed without it.

In our example of a symmetric ROC curve, we found that CD4[+] T[EM] cells and 9 subordinate populations, as well as 6 other populations, were significant discriminators in both the unrestricted and restricted datasets (Fig. 6a). Four further populations were significant only in the restricted dataset. In the second example, which resulted in a right-skewed ROC curve, we found CD4[+] T[EM] cells, two subpopulations and three CD4[−] naïve T cell (T[naive]) subordinates with a significant restricted p-value, whereas the corresponding unrestricted permutation p-value was insignificant (Fig. 6b). In the third example, which resulted in a left-skewed ROC curve, we found that CD4[+] T[EM], 4 subordinates and 8 other subsets had a significant restricted p-value, but were insignificant in the unrestricted dataset (Fig. 6c). In the fourth example, CD4[+] T[EM] cells had an optimal restriction permutation p-value = 0.002, but were insignificant in the unrestricted dataset (Fig. 6d). Hence,

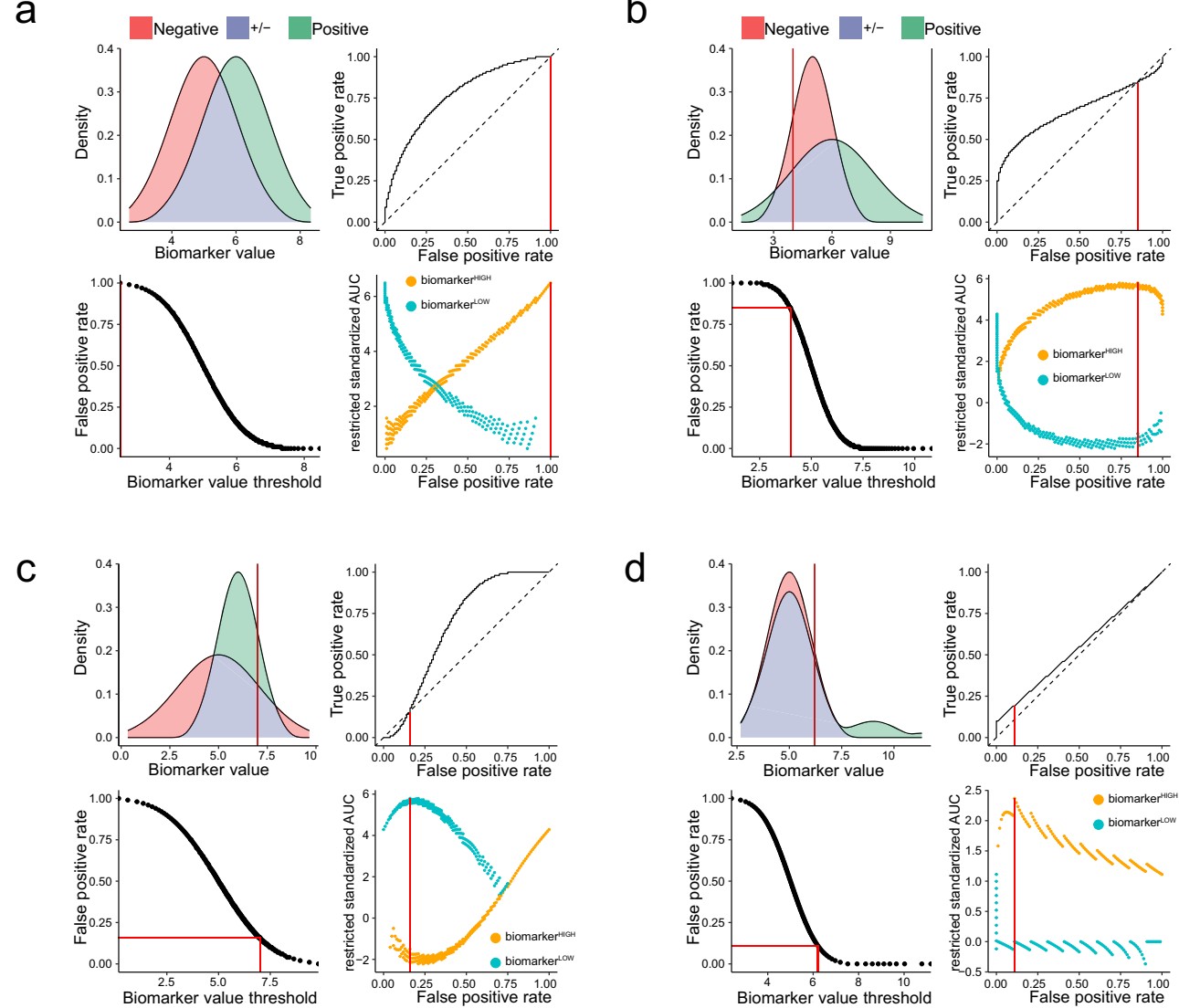

**Fig. 3 | Optimal restriction of two-class distributions results in asymmetric ROC curves.** We present four simulated examples of biomarker distributions in two classes, which are intended to represent sets of patients with different clinical outcomes. The distribution of values from the positive (i.e. diseased) class are coloured green and values from the negative (i.e. control) class are coloured red; the overlapping density areas are coloured purple. For each example, we present the following: a plot of positive and negative class densities; the complete ROC curve; a plot of biomarker values against FPR; a plot of rzAUC calculated for biomarker[HIGH] (orange) and biomarker[LOW] (blue) samples at all FPR values. In each plot, red lines indicate the optimal restriction as a biomarker value or FPR value. **a** A simulated example of a symmetric ROC curve from 100 negative $\mathcal{N}(5,1)$ and 100 positive $\mathcal{N}(6,1)$ samples. **b** A simulated example of a right-skewed ROC curve from 100 negative $\mathcal{N}(5,1)$ and 100 positive $\mathcal{N}(6,2)$ samples. **c** A simulated example of a left-skewed ROC curve from 100 negative $\mathcal{N}(5,2)$ and 100 positive $\mathcal{N}(6,1)$ samples. **d** Results for a right-skewed ROC curve from 100 negative $\mathcal{N}(5,1)$ samples and 100 positive samples from a bimodally distributed positive population. In this example, the positive population comprises 10% cases with elevated biomarker expression $\mathcal{N}(9,1)$ and 90% cases with unaltered biomarker expression $\mathcal{N}(5,1)$.

dataset restriction enables the discovery of disease biomarkers which would otherwise be disregarded in synthesised flow cytometry datasets.

## Dataset restriction discovers valid irAE biomarkers

Having qualified our restriction method using synthesised datasets, we next applied it to real clinical data. In previous work, we investigated pre-treatment peripheral blood samples from 110 patients with advanced melanoma who received Ipi-Nivo therapy[13]. Using conventional methods, we found no significant biomarker after correcting for multiple comparisons; therefore, we asked whether our restriction method could reveal biomarkers of hepatitis or colitis risk in the same dataset (Fig. 6e, f). No biomarkers of colitis survived correction for multiple comparisons after restriction (Fig. 6e and Supplementary

Fig. 8). However, in predicting hepatitis, our restriction method returned 7 significant biomarkers with a permutation p-value < 0.05 (Fig. 6f). After correction for multiple testing, 4 of these 7 hepatitis biomarkers remained significant with an FDR < 0.05. By contrast, no biomarker identified from the unrestricted dataset returned a significant permutation p-value after correction for multiple testing. Thus, our restriction method returned significant disease-associated biomarkers in a real-world dataset, which were not found using the unrestricted dataset.

Using our restriction method, we identified CD27[+] CD28[+] CD4[+] T[EM] cell frequency relative to CD4[+] in blood as a biomarker of hepatitis risk after dataset restriction. To illustrate the potential utility of restricted biomarkers, we compared the performance of CD27[+] CD28[+] CD4[+] T[EM] frequency as a biomarker of hepatitis risk in our unrestricted

## BOX 2

# Cytometry Simulation

**I. Parameter estimation:**

**Input:**

- Compensated, asinh-transformed matrices $\boldsymbol{X}^{(i)} \in \mathbb{R}^{n_i \times m}$ with $n_i$ rows (cells) and $m$ columns (measured cell antigens) for $N$ samples $i$ in $\{1, \ldots, N\}$.
- Fixed hierarchical gating $g(\cdot)$ assigning one of $K$ leaf cell populations to each cell $\boldsymbol{X}^{(i)}_{c,\cdot} \in \mathbb{R}^m$ according to $m$ cell antigens: $g(\boldsymbol{X}^{(i)}_{c,\cdot}) \in \{1, \ldots, K\}$.

**Output:** Parameters $\hat{\alpha}_k, \hat{\boldsymbol{\mu}}_k, \hat{\boldsymbol{\Sigma}}_k$ for $k \in \{1, \ldots, K\}$ of a Dirichlet process Gaussian mixture model.

**Algorithm:**

1. For all samples $i$:
   1a. Assign leaf cell populations to all cells $c$
   $$\text{pop}^{(i)}_c = g(\boldsymbol{X}^{(i)}_{c,\cdot}).$$

   1b. For all $k$ populations calculate population proportions
   $$p^{(i)}_k = \frac{1}{n_i} \sum_{c=1}^{n_i} \delta_{k,\text{pop}^{(i)}_c}.$$

2. Account for empty populations by
   $$p^{(i)''}_k = \frac{p^{(i)'}_k}{\sum_{\ell=1}^{K} p^{(i)'}_\ell},$$
   with
   $$p^{(i)'}_k = p^{(i)}_k + 0.001 \cdot \min_{\ell,j}(\{p^{(j)}_\ell | p^{(j)}_\ell > 0\}).$$

3. Iterative maximum likelihood estimation of the Dirichlet distribution with $K$ parameters $\hat{\boldsymbol{\alpha}}$ using $p''$ results in
   $$\text{Dir}(\hat{\alpha}_1, \ldots, \hat{\alpha}_K).$$

4. For each population $k$ estimate mean $\hat{\boldsymbol{\mu}}_k \in \mathbb{R}^m$ and covariance matrix $\hat{\boldsymbol{\Sigma}}_k \in \mathbb{R}^{m \times m}$ using all cells $c$ from all samples $i$ with $\text{pop}^{(i)}_c = k$ to establish a multivariate normal distribution
   $$\mathcal{N}(\hat{\boldsymbol{\mu}}_k, \hat{\boldsymbol{\Sigma}}_k).$$

**II. Parameter adjustment for disease effects:**

**Input:**

- Dirichlet parameters $\alpha_k$ for $k \in \{1, \ldots, K\}$.
- Target mean percentage $t \in (0, 1)$ for a given subset of leaf populations $A \subset \{1, \ldots, K\}$.

**Output:** Adjusted Dirichlet parameters $\alpha'_k$ such that the proportion of cells from $A$ is $t$ and $\sum_{k=1}^K \alpha'_k = \sum_{k=1}^K \alpha_k$.

**Algorithm:**

1. Define the complement of $A$ as $\bar{A} := \{1, \ldots, K\} \setminus A$ and precisions $s$ as
   $$s_A := \sum_{k \in A} \alpha_k, \quad s_{\bar{A}} := \sum_{k \in \bar{A}} \alpha_k, \quad s := s_A + s_{\bar{A}}.$$

2. Calculate modified parameters
   $$\alpha'_k := \begin{cases} \alpha_k \frac{ts}{s_A}, & \text{if } k \in A, \\ \alpha_k \frac{(1-t)s}{s_{\bar{A}}}, & \text{else}, \end{cases}$$
   which ensures that
   $$\sum_{k \in A} \alpha'_k = ts \quad \text{and} \quad \sum_{k \in \bar{A}} \alpha'_k = (1-t)s.$$

**III. Sample simulation:**

**Input:**

- Target number of cells $C$.
- Parameters $\alpha_k \in \mathbb{R}_{\geq 0}, \boldsymbol{\mu}_k \in \mathbb{R}^m, \boldsymbol{\Sigma}_k \in \mathbb{R}^{m \times m}$ for $k \in \{1, \ldots, K\}$.

**Output:** Simulated matrix $\boldsymbol{X} \in \mathbb{R}^{C \times m}$ of $C$ cells with $m$ cell antigens.

**Algorithm:**

1. Sample a proportion vector $\boldsymbol{p}$ from the Dirichlet distribution
   $$\boldsymbol{p} \sim \text{Dir}(\alpha_1, \ldots, \alpha_K).$$

2. For each leaf population $k$, sample $C \cdot p_k$ cells from $\mathcal{N}(\boldsymbol{\mu}_k, \boldsymbol{\Sigma}_k)$.

and restricted datasets (Fig. 7). The discriminatory cutoff for patient classification, defined by the Youden index, was the same for both the restricted and unrestricted datasets, such that samples with more than 9.56% of CD27$^+$ CD28$^+$ CD4$^+$ T$_{\text{EM}}$ relative to CD4$^+$ are predicted to be hepatitis positive. Accordingly, using the unrestricted dataset, CD27$^+$ CD28$^+$ CD4$^+$ T$_{\text{EM}}$ (%) correctly predicted the incidence of hepatitis in 74 of 110 patients. The unrestricted cell frequency had a sensitivity (TPR) of 45.8% and a specificity (true negative rate, TNR) of 83.9%. The positive predictive value (PPV) was 68.8% and the negative predictive value (NPV) was 66.7%. Our method of restricting biomarkers to their informative ranges implies that some samples should be considered unclassifiable. In this example, 58 of 110 patients were unclassifiable. The incidence of hepatitis was correctly predicted in 40 of 52 classifiable samples. The restricted cell frequency had a sensitivity of 91.7%

and a specificity of 64.3%. The positive predictive value was 68.8% and the negative predictive value was 90%.

To explore the applicability of restriction to other sources and types of immunological data, we applied our method to repurposed datasets published by other groups, including proteomic[39], mass cytometric[16], microbiomic[40] and transcriptomic[41] studies. New univariate markers were discovered in each case (Supplementary Figs. 9–13).

**Multivariate analysis of restricted data predicts hepatitis**

Although our restriction method leads to discarding samples as unclassifiable according to any particular biomarker, we found that different biomarkers define noncongruent sets of classifiable samples (Fig. 8a). This led us to investigate whether using restricted datasets

could improve the predictive performance of multivariate models. First, we built a random forest model[42] using all 84 reported T cell subset frequencies from the unrestricted training dataset of 110 patients (Fig. 8b). When this model was applied to an independent, prospective validation set of 30 patients, the resulting predictions were inaccurate (correct classification rate = 56.7% vs. 53.3% under the no-information model).

By contrast, we observed a significant improvement in predictive performance using the restricted dataset to train our random forest. In this approach, to avoid "double-dipping," we exclusively used information from the training set to establish restriction values and train the random forest. To leverage information from our restriction method, we assigned a value of −1 to restricted samples across all 84 biomarkers. When restriction values and our predictive model were applied to the validation set, the resulting predictions were significant (Fisher's Exact p-value = 0.026) and had a correct classification rate of 73.3%. 12 of 16 predictions of hepatitis were correct (PPV = 75%) and 10 of 14 negative predictions were correct (NPV = 71.4%). Hence, in principle, dataset restriction can improve the training and performance of multivariate predictive models based upon real-world data.

## Generalisation of restriction values across public datasets

Next, we investigated whether the performance of multivariate models built with other data types from external sources could be improved through dataset restriction. We applied our method to transcriptomic data from 921 samples aggregated from 10 published studies that examined clinical response to ICI therapy across a variety of cancers[41]. Cases from 5 studies were split into training ($n = 618$) and validation ($n = 154$) sets. The test set ($n = 149$) was compiled from 5 separate studies. Univariate analysis of the restricted training set revealed 19 genes missed by global analysis, including RAC1 and CEACAM6 (Supplementary Fig. 13). We then constructed four multivariate random forests with or without restriction, and with or without random forest hyperparameter optimisation. Crucially, only information from the training set was used to set restriction values and to train and optimise our models. Restriction improved predictive performance in the validation and independent test datasets (Supplementary Fig. 14).

Finally, we demonstrated that restriction preprocessing generally improved multivariate random forest performance regardless of the split into training, validation and test set (Supplementary Fig. 15). We repeatedly randomised the 921 samples into 70% training and 30% test samples to construct multivariate random forests with or without restriction. In the majority of 750 random splits, we observed an increase in the AUC on the test set after restriction. This suggests that dataset restriction will typically improve predictive models using immunological data. Furthermore, it implies that restriction values can be generalised across independent datasets.

## Discussion

Immunological diseases are often heterogeneous in clinical presentation and severity, reflecting the variability of their underlying immunopathologies[15]. It follows, we argue, that immune disease-associated biomarkers typically exhibit greater variability among diseased patients than unaffected individuals. This general proposition was broadly corroborated by our real-world examples of patient groups who were prone to immunotherapy-related complications. Unequal dispersion of biomarker distribution between patient classes affects our ability to identify biomarkers with discriminatory capacity over a certain range of biomarker values. To solve this biological problem, we introduced dataset restriction as a biomarker discovery tool. In artificial and real-world examples, dataset restriction enabled us to find discriminatory biomarkers that were undetected by conventional measures. Moreover, we showed that dataset restriction improves the performance of multivariate predictive models. Our work formalises a new way of evaluating diagnostic results –

specifically, that certain biomarkers can only be usefully interpreted over a restricted range of values, and that samples with values outside this range should be considered as unclassifiable.

Flow cytometry is a powerful method for interrogating the phenotype of many single cells within a heterogeneous mixture. This technique allowed us to estimate the relative numbers of accurately defined leukocyte subsets in peripheral blood samples, including T cell subsets, which are direct targets of Ipilimumab (anti-CTLA-4) and Nivolumab (anti-PD-1) therapy[43]. Although flow cytometry generates rich and immunologically interpretable data, it has two key limitations – namely, that blood leukocyte frequencies vary within a narrow dynamic range, and that higher-order cell antigen combinations may define rare cell subsets[44,45]. Small disease-related changes in biomarkers are problematic because substantially overlapping biomarker distributions with unequal variability lead to exaggerated skewness of ROC curves, implying unequal informativeness of those biomarkers across their measurable ranges. Rare cell subsets are problematic because our estimates of their frequency are less reliable[46]. Crucially, dataset restriction helps to overcome the special difficulties of correctly interpreting flow cytometry data by limiting biomarker values to a range in which the signal-to-noise ratio is increased relative to the full range. Consequently, we reduce the likelihood of false positive or false negative classification at the cost of discarding some samples as unclassifiable.

We created restrictedROC[36], an R-package that calculates restricted standardised AUC scores. The rzAUC is returned together with a restriction value that delimits the biomarker's optimally informative range. This builds upon earlier ideas about partial AUCs, which were introduced to account for imposed restrictions that capped true and false positive rates[47–49]. Imposed restrictions usually come from domain knowledge; for instance, tests with a high false positive rate are inappropriate for expensive diagnostic screening applications, whereas tests with a high false negative rate are inappropriate when a life-saving treatment is available[50]. McClish introduced a "standardisation" for partial AUCs for a given range of false positive rates, such that a randomly selected positive sample has a higher value than a randomly selected negative sample conditional upon the negative sample arising from the false positive range[51]. In our method, we introduced a scaling factor for the two-way partial AUC[50] resulting in the restricted AUC (rAUC). With this scaling factor, the rAUC becomes the probability that a randomly selected positive sample has a higher value than a randomly selected negative sample conditional upon both samples arising from a range spanned by a minimum true positive rate and a maximal false positive rate. The restricted standardised AUC (rzAUC) then takes into account both the rAUC and the number of samples in the biomarker^HIGH or biomarker^LOW range leveraging the equivalence between AUC and Mann-Whitney U test[33].

We further developed our method to determine the optimal range of biomarker values that correctly classifies samples. Specifically, we optimise a restriction that either includes samples with higher biomarker values (biomarker^HIGH) or lower biomarker values (biomarker^LOW) and has the highest possible absolute rzAUC. The rzAUC can be directly compared within one dataset but depends on the total number of samples. By calculating permutation p-values[38] for the rzAUC, we remove this dependence and attribute significance values.

There are alternative ways of describing the geometric symmetry of ROC curves apart from graphical skewness. Left-skewed ROC curves are also described as True Negative Proportion (TNP)-asymmetric and right-skewed ROC curves as True Positive Proportion (TPP)-asymmetric. These asymmetries can be defined by Kullback-Leibler (KL-) divergences[52]. Therefore, KL-divergence could be used to assess whether restriction should be applied to a given biomarker; however, in the case of symmetric ROC curves, our restriction keeps all samples, so such preselection of biomarkers is unnecessary. Of note, excluding

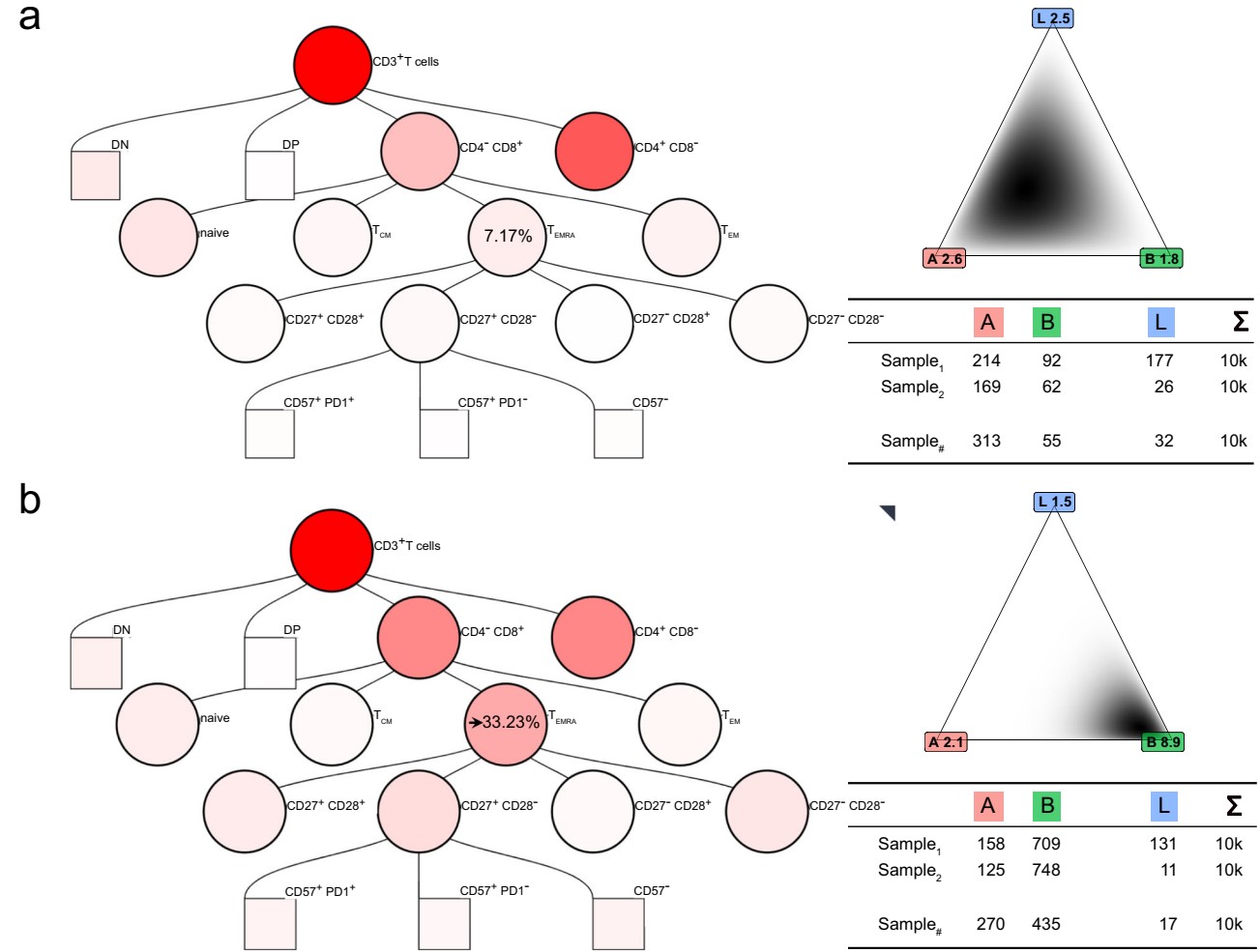

**Fig. 4 | Realistically synthesising flow cytometry data from two class distributions.** Various applications of our synthetic flow cytometry data depend upon generating samples with differences in cell subset distributions. Here, we provide an example of increasing the proportion of CD8$^+$ T$_{EMRA}$ cells in synthetic samples from a baseline value of 7.17% in the negative class to an altered value of 33.23% in the positive class. The intensity of red shading in the gating trees illustrates this change in CD8$^+$ T$_{EMRA}$ cells and contingent changes in other populations. **a** Gating tree with 7.17% CD8$^+$ T$_{EMRA}$ cells and the density for three example gates: A, B and L. The cell count table for three samples drawn from this distribution is shown. **b** Gating tree with 33.23% CD8$^+$ T$_{EMRA}$ cells and the density for three example gates: A, B and L. The cell count table for three samples drawn from this distribution is shown. Of special note, percentages of cells in all other gates also changed according to the Dirichlet distribution, leading to changes in simulated cell counts across all leaf gates.

samples to minimise KL-divergence is not the equivalent of dataset restriction.

In principle, dataset restriction can be applied to optimise any biomarker range. However, following from our immunological rationale, restricting the upper or lower range is especially applicable in clinical diagnostics. For completeness of our discussion, we can imagine a biomarker with both uninformative biomarker$^{HIGH}$ and biomarker$^{LOW}$ values (ie. where only mid-range values are informative) that might only be discovered by applying our restriction method twice in succession.

To validate our restriction method, we developed a method for synthesising realistic flow cytometry data with class-related effects. Because no generative method previously existed, our approach represents a significant contribution to cytometry analysis, particularly for benchmarking of diagnostic flow cytometry algorithms, sample size calculations or data augmentation. Our method uses an expert-given hierarchical gating strategy, where the proportions of cells per gate are described with a Dirichlet distribution. Within each terminal (leaf) gate, the cells are described using a normal distribution. Thus, we effectively created a Gaussian mixture distribution with the number of components defined by the number of terminal gates. In cytometry,

(Gaussian) mixture models are an established method for unsupervised cell population identification[53,54]. In principle, these earlier approaches could be used to simulate cells from estimated distributions, although their focus was labelling existing cells rather than creating artificial ones.

Synthesising data by Gaussian mixture models allows for the creation of many complex data distributions but has two limitations. First, the choice of a multivariate Gaussian distribution for cell antigen expressions at each leaf gate is simple and effective but could be improved by multivariate skew t-distributions[55] to better describe the outlier-heavy nature of flow cytometry cell measurements. Second, the Dirichlet distribution incorporates modifications in the proportion of any cell population by changing all other proportions; notably, this is a simplification that doesn't incorporate biological dependencies between cell subset frequencies. Despite these limitations, our generative model is suitable for its application in this work – namely, testing the performance of restriction in simulated datasets with a realistic level of noise.

In this study, we present idealised flow cytometry data generated under highly standardised conditions using only two very closely aligned instruments. Consequently, these data do not

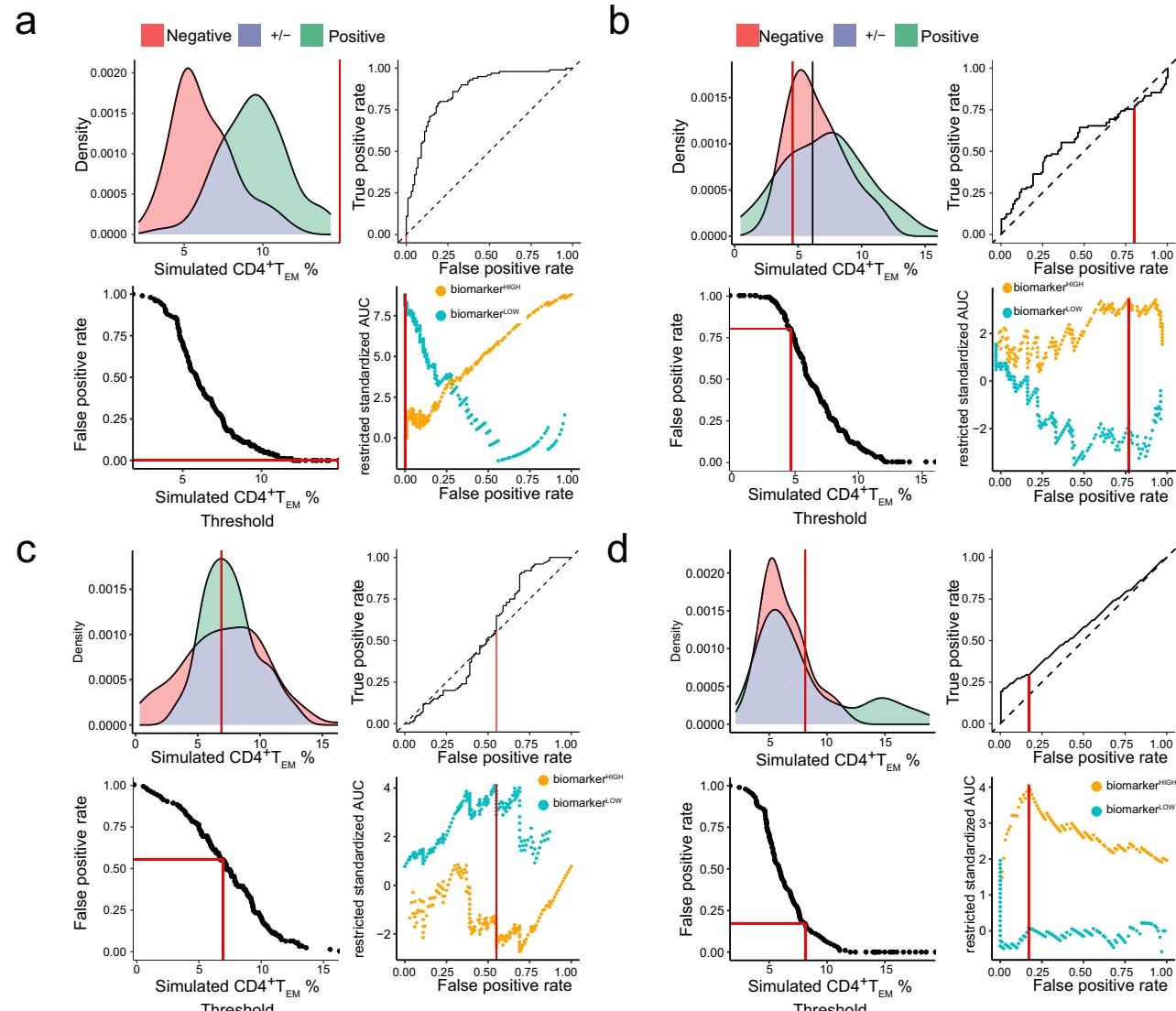

**Fig. 5 | Restriction of synthesised flow cytometry datasets.** We present examples of biomarker distributions in two classes, which are intended to represent sets of patients with different clinical outcomes. The distribution of values from positive (i.e. diseased) class are coloured green and values from negative (i.e. control) class are coloured red; overlapping density areas are coloured purple. For each example, we present the following: a plot of positive and negative class densities; the complete ROC curve; a plot of biomarker values against FPR; a plot of rzAUC calculated for biomarker$^{HIGH}$ (orange) and biomarker$^{LOW}$ (blue) samples at all FPR values. In each plot, red lines indicate the optimal restriction as a biomarker value or FPR

value. **a** A synthetic example of a symmetrical ROC curve from 100 negative $\mathcal{N}(7.7,1)$ and 100 positive $\mathcal{N}(10.7,1)$ samples. **b** A synthetic example of a right-skewed ROC curve from 100 negative $\mathcal{N}(7.7,1)$ and 100 positive $\mathcal{N}(8.7,3)$ samples. **c** A synthetic example of a left-skewed ROC curve from 100 negative $\mathcal{N}(7.7,3)$ and 100 positive $\mathcal{N}(8.7,1)$ samples. **d** Results for a synthetic right-skewed ROC curve from 100 negative $\mathcal{N}(7.7,1)$ samples and 100 positive samples from a bimodally distributed positive population. In this example, the positive population comprises 20% cases with elevated biomarker expression $\mathcal{N}(16.7,1)$ and 80% cases with unaltered biomarker expression $\mathcal{N}(7.7,1)$.

reflect the typical quality of clinical flow cytometry measurements, especially when multiple operators, instruments and site-to-site differences in protocols contribute to variability. Using multicolour flow cytometry for clinical classification tasks is substantially complicated by real-world shifts and drifts in assay performance. Progress is being made in overcoming these challenges in three ways – namely, standardisation[56,57], calibration[58,59] and normalisation[60,61]. However, the goal of a clinical decision-making tool that can be applied to flow cytometry data from any laboratory without relying upon measurements of paired samples, exchange of external reference material or sharing patient-level data has not been realised. Neural networks[45] offer a promising solution for improving transferability of predictive models that use flow cytometry data. In the future, our approach to

simulating realistic flow cytometry data could allow the pre-training of neural networks, reducing their sensitivity to technical effects.

Restricting biomarkers to an informative range is important because it improves classification performance. We emphasise that classification cutoffs and restriction values are different concepts. Classification cutoffs, such as the Youden index[62], divide a sample set into predicted positive and predicted negative classes. By contrast, restriction divides a sample set into classifiable and unclassifiable samples. In the context of individualised patient care, it might seem unhelpful to label samples as unclassifiable. On the contrary, we argue that the clinical utility of a predictive biomarker improves if its certainty is high, even if it only works in a small subset of patients. Consider a disease-related biomarker giving a right-skewed ROC curve:

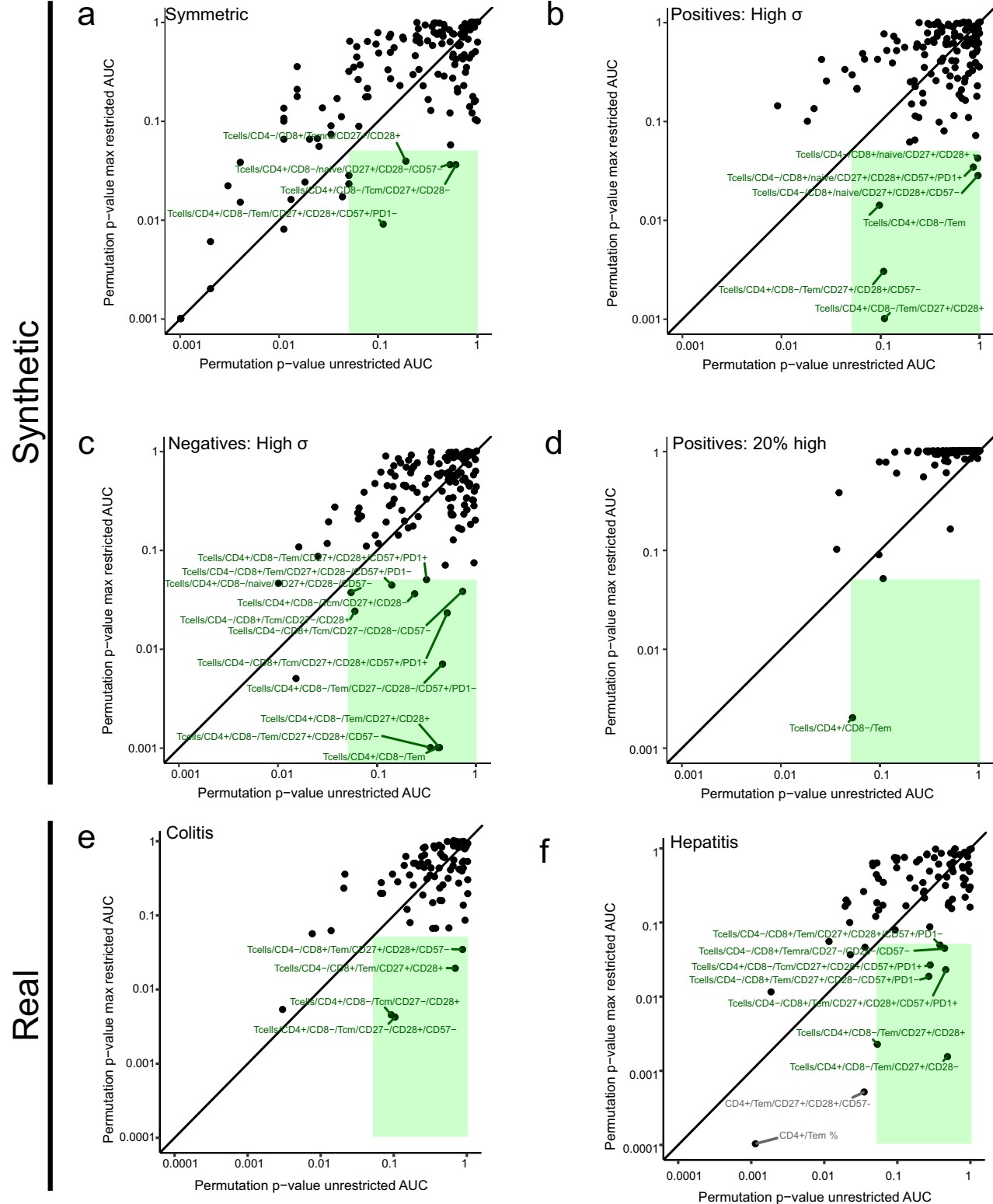

Conventional approaches return a reliable positive classification and an unreliable negative classification; in contrast, our restriction method returns a reliable positive classification, a reliable negative classification and a set of unclassifiable samples, which do not necessarily have the most negative values. Of note, the discriminatory cutoff determined by the Youden index is often the same after restriction, but changes in some cases. When interpreting a single biomarker, our restriction method improves

either the positive or the negative predictive value, so improves certainty of our predictions.

Our method may concern some clinicians, who will legitimately ask about unclassifiable patients[63]. Here, we provide an answer by building an informative and prospectively validated random forest model after replacing all restricted values with a constant outside the informative range. Consequently, we force each tree of the random forest to select discriminatory cutoffs within the informative range or a

**Fig. 6 | Restriction augments biomarker discovery in synthetic and real-world flow cytometry datasets.** Using synthetic and real-world datasets, we demonstrate that restriction augments the discovery of biomarkers with limited informative ranges. In each plot, the x-axis shows permutation p-values for the AUC of complete ROC curves for every gated cell population. The y-axis shows permutation p-values for the AUC of optimally restricted ROC curves. Points within the green-shaded rectangles represent cell subsets whose p-values derived from unrestricted data are not significant (p ≥ 0.05), but p-values derived from optimally restricted data are significant (p < 0.05). **a** Permutation p-values from synthetic samples in which a disease-related effect was introduced into CD4$^+$ T$_{EM}$ resulting in a symmetric ROC curve. 100 samples in the negative $\mathcal{N}(7.7,1)$ class and 100 samples in the positive $\mathcal{N}(10.7,1)$ class were generated. **b** Permutation p-values from synthetic samples in which a disease-related effect was introduced into CD4$^+$ T$_{EM}$ resulting in a right-skewed ROC curve. 100 samples in the negative $\mathcal{N}(7.7,1)$ class and 100 samples in the positive $\mathcal{N}(8.7,3)$ class were generated. **c** Permutation p-values from synthetic samples in which a disease-related effect was introduced into CD4$^+$ T$_{EM}$ resulting in a left-skewed ROC curve. 100 samples in the negative $\mathcal{N}(7.7,3)$ class and 100 samples in the positive $\mathcal{N}(8.7,1)$ class were generated. **d** Permutation p-values from synthetic samples in which a disease-related effect was introduced into CD4$^+$ T$_{EM}$ resulting in a right-skewed ROC curve. 100 samples in the negative $\mathcal{N}(7.7,1)$ class and 100 samples from a bimodally distributed positive class were generated. In this example, the positive population comprises 20% cases with elevated biomarker expression $\mathcal{N}(16.7,1)$ and 80% cases with unaltered biomarker expression $\mathcal{N}(7.7,1)$. **e** Permutation p-values from a training set of real-world clinical flow cytometry samples ($n = 110$). 84 biomarkers were selected where ≥ 10% of samples had more than 10 counts. Dataset restriction reveals 4 previously undescribed biomarkers of treatment-related colitis risk in metastatic melanoma patients receiving Ipi-Nivo therapy. **f** Permutation p-values from a training set of real-world clinical flow cytometry samples ($n = 110$). 84 biomarkers were selected where ≥ 10% of samples had more than 10 counts. Dataset restriction reveals 7 previously undescribed biomarkers of treatment-related hepatitis risk in metastatic melanoma patients receiving Ipi-Nivo therapy.

cutoff between the classifiable and unclassifiable regions for each biomarker. More sophisticated methods may be developed in the future, but our experimentally validated random forests are proof of the principle that differently restricted biomarkers can be usefully combined in multivariate models.

To demonstrate the potential clinical utility of dataset restriction, we applied our method to the clinically significant problem of immune-related adverse events following combined immunotherapy. In univariate analyses, dataset restriction identified new biomarkers associated with ICI-related hepatitis, including CD27$^+$ CD28$^+$ CD4$^+$ T$_{EM}$ cells, that were not returned by conventional methods. Of clinical importance, dataset restriction increased NPV without compromising PPV. Combining many restricted biomarkers into a random forest model generated an informative model, whereas training on unrestricted data from the same set of 110 samples returned no valid models. To validate our predictive model, we assessed its performance in an independent, prospectively collected set of 30 samples, where it returned significant predictions that were superior to the performance of any single biomarker alone. Beyond the scope of this article, such multivariate models could be extended to include biomarkers from multiple flow cytometry panels or other patient-related information, such as age, sex or clinical chemistry results. In support of this claim, restriction improved prediction of clinical responses in ICI-treated patients from public transcriptomic data aggregated from many independent studies.

Clinical manifestations of immune disease are often heterogeneous. This is certainly true of irAE after immunotherapy, which vary greatly in severity, time-of-onset, clinical features and response to treatment[28]. Further, there is increasing evidence that multiple immune aetiologies lead to common clinical presentations, such as colitis[64], myositis[65] or hepatitis[66]. This heterogeneity connotes individual genetic predisposition[67,68], environmental factors[69,70] and past immunological challenges[35]. In particular, we now recognise the contribution of previous viral infections in preconditioning towards adverse reactions. An unanticipated consequence of dataset restriction is that disease biomarkers with a bimodal distribution in the positive class, such as might arise from multiple aetiologies, are findable. Excitingly, combining biomarkers from a restricted dataset into multivariate models should, in principle, enable predictions about diseases with multiple aetiotypes – a situation where conventional biostatistical methods are unsuitable. Extending this idea of dataset restriction as a way of classifying samples with intraclass heterogeneity to unsupervised methods, such as PCA or clustering, could aid discovery of previously unknown patient subsets.

The core insight from our work is that biomarkers of immune disease are often more variably expressed in affected populations than in healthy comparators. Many factors might contribute to this higher variability within diseased groups, such as individual patients' age, sex, genetics, comorbidities, concurrent therapies, stage of disease at sampling, or alternative aetiopathologies. Our example of CMV-associated expansion of CD4$^+$ T$_{EM}$ cells predisposing to ICI-related hepatitis illustrates the influence of unanticipated variables over biomarker performance in heterogeneous populations. When relevant subgroups within classes are known, random effects models[71] are useful in controlling for unobserved heterogeneity by introducing subgroup-specific weightings. Notably, univariate biomarkers in a discovery study are likely influenced by different sets of subject-specific factors; therefore, each biomarker must be modelled separately. In contrast, dataset restriction requires no prior knowledge about intraclass heterogeneity and is not itself a method for defining subgroups or finding latent variables. Whether splitting datasets into informative and uninformative sample sets enriches for relevant, but unknown subgroups within classes is yet unexplored. Another crucial aspect of dataset restriction is its use as a preprocessing step in establishing multivariate biomarkers that are agnostic about intraclass heterogeneity. We are not aware of random effects models currently being applied in this context.

In summary, clinical biomarkers that can only be interpreted over a restricted range are inherently likely in immune diseases. Where classical methods fail, dataset restriction often solves the problem of discovering and interpreting such biomarkers. Our approach is not limited to prospective data, but can also be used retrospectively to find new biomarkers or improve existing ones. Dataset restriction was developed here to analyse flow cytometry data; however, it is directly applicable to any sample classification problem. We hope others will apply our method to existing datasets.

## Methods
### Collection of clinical information
Locally generated data from three sources were used in this study: (1) a training set ($n = 48$) from a cohort of healthy humans used to develop our flow cytometry data simulations; (2) a previously reported training set ($n = 110$) from patients with advanced melanoma used for biomarker discovery[13]; and (3) a new prospective validation set ($n = 30$) from patients with advanced melanoma. Whole blood was collected from healthy thrombocyte donors with approval from the Ethics Committee of the University of Regensburg (approval 22-2780-101). All donors gave full, written consent to sample and data collection.

Clinical samples for the biomarker training and validation sets were collected within a single-centre, non-interventional study[72], which was conducted in accordance with the Declaration of Helsinki and all applicable German and European laws and ethical standards. This observational study was authorised by the Ethics Committee of the University of Regensburg (approval 16−101-0125) and registered

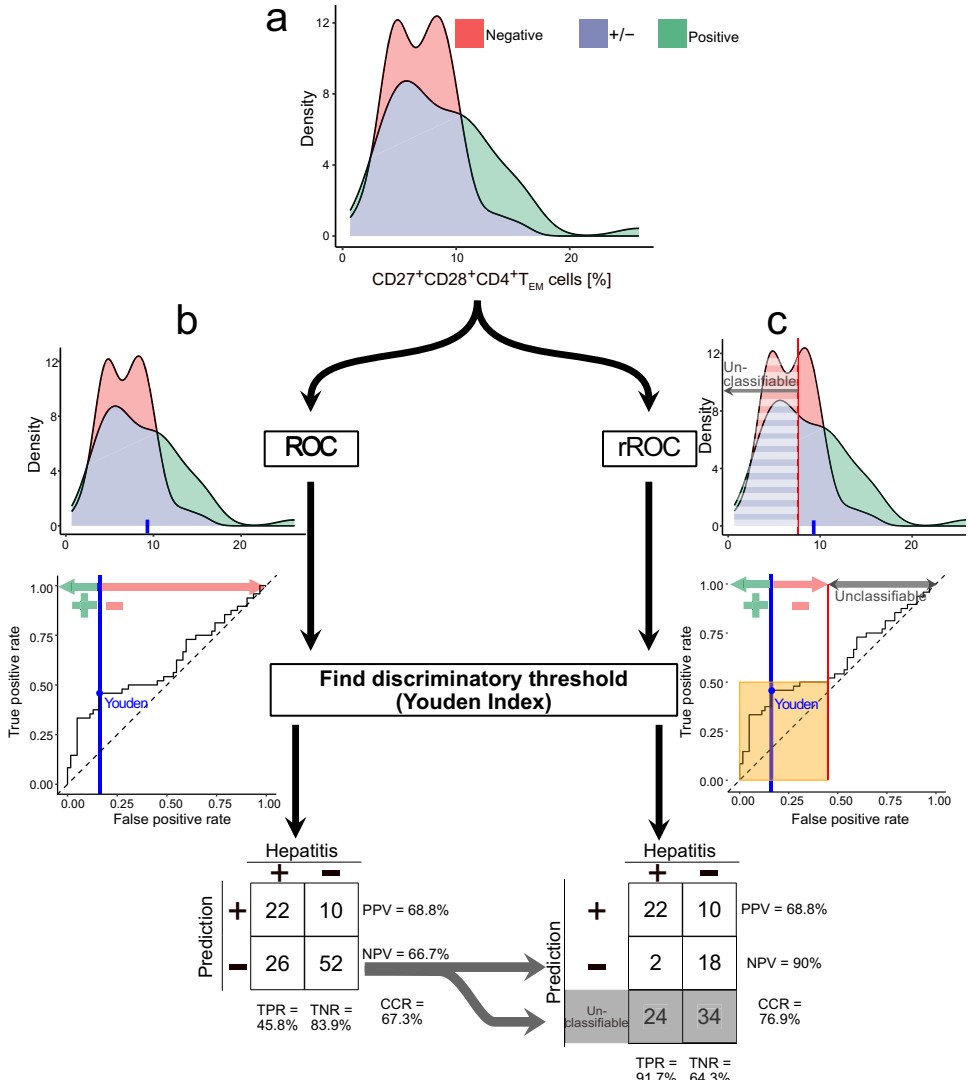

**Fig. 7 | Clinical interpretation of restricted biomarkers in predicting disease.** Our method of dataset restriction leads to counterintuitive clinical interpretations of biomarker values. This is illustrated by our discovery of CD27$^+$ CD28$^+$ CD4$^+$ T$_{EM}$ cells as a univariate biomarker of hepatitis risk after immunotherapy. Here, we illustrate the conventional evaluation of biomarker performance across all samples with evaluation of biomarker performance in a restricted dataset. **a** Densities of CD27$^+$ CD28$^+$ CD4$^+$ T$_{EM}$ cells in all samples from patients with metastatic melanoma who developed hepatitis ($n = 48$) or did not ($n = 62$) after starting Ipi-Nivo therapy. **b** Following the classical approach of determining a classification cutoff for CD27$^+$ CD28$^+$ CD4$^+$ T$_{EM}$ frequency relative to CD4$^+$ T cells using the Youden Index, we predict hepatitis if > 9.62% and then assess the correct classification rate (CCR), negative predictive value (NPV), positive predictive value (PPV), sensitivity (or true positive rate, TPR) and specificity (or true negative rate, TNR) for all samples. **c** Our restriction method is predicated on there being a range of values over which a biomarker provides no discriminatory information. Optimally restricting CD27$^+$ CD28$^+$ CD4$^+$ T$_{EM}$ cell values leads us to discard 58 of 110 samples as "unclassifiable." For the remaining 42 samples where CD27$^+$ CD28$^+$ CD4$^+$ T$_{EM}$ frequency relative to CD4$^+$ T cells > 7.62%, we determine a classification cutoff using the Youden Index, again predicting hepatitis if > 9.62%. Accordingly, we obtain a confusion table with CCR = 76.9%, specificity = 64.3% sensitivity = 91.7%, PPV = 68.8% and NPV = 90% across the classifiable samples.

with clinicaltrials.gov (NCT04158544). Blood samples were obtained from patients with Stage III/IV melanoma under the care of the Department of Dermatology at University Hospital Regensburg (UKR). Eligible patients were consecutively recruited without stratification or matching. All participants gave full, informed written consent. For the training set, the first reported case was recruited in OCT-2016 and the last reported case was recruited in JUN-2021. For the prospective validation set, the first reported case was recruited in JUN-2021 and the last reported case was recruited in JAN-2023 (Supplementary Table 1). All study participants received standard-of-care treatment according to local guidelines. Specifically, patients with unresectable metastatic disease who received first- or second-line checkpoint inhibitor therapy were initially treated with Nivolumab (αPD-1; 1 mg/kg; Bristol-Myers Squibb) plus Ipilimumab (αCTLA-4; 3 mg/kg; Bristol-Myers Squibb) for

up to four cycles at 3-week intervals. Thereafter, patients received 480 mg Nivolumab monotherapy at 4-week intervals.

## Diagnosis and grading of clinical outcomes

All irAE were evaluated by an expert Dermatological Oncologist. ICI-related hepatitis was diagnosed when: (i) glutamic oxaloacetic transaminase (GOT), glutamic pyruvic transaminase (GPT), γ-GT or total bilirubin substantially deviated from pretreatment values; (ii) this change was not attributable to other causes, such as co-medication or viral disease; and (iii) liver injury was sufficiently severe that ICI therapy was suspended or stopped, or immunosuppression was started. Colitis was diagnosed when increased stool frequency or loose consistency was accompanied by abdominal discomfort, leading to suspension or cessation of ICI therapy and introduction of immunosuppressive

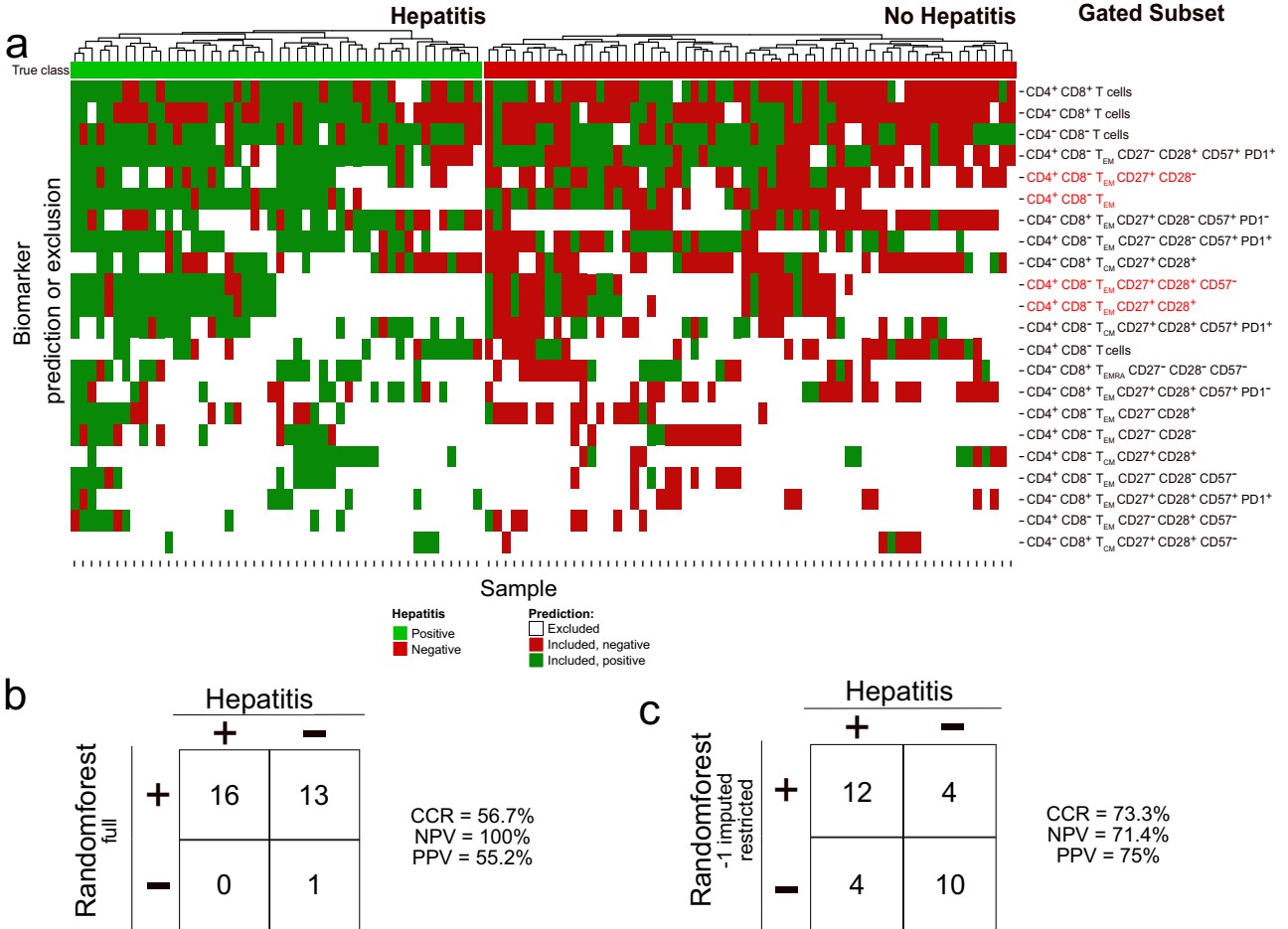

**Fig. 8 | Significant biomarkers according to classical and our analysis predicting hepatitis. a** Heatmap showing significant biomarkers of hepatitis risk after Ipi-Nivo therapy based upon permutation p-values for unrestricted and restricted AUC. After correction for multiple testing, only 4 biomarkers remained significant, all discovered by our restriction method and indicated by red text. Each further column reflects one sample, each row a biomarker. The samples are grouped into patients who did (green) or did not (red) develop treatment-related hepatitis, shown in the very first row. The main matrix consists of three values: Those excluded according to restriction (white); those included and predicted positive (dark green); and those included and predicted negative (dark red). Columns were clustered, rows in increasing order according to the number of excluded samples. **b** Random forest predictions and performances on the prospective validation cohort ($n = 30$) trained on all, not only the significant, unrestricted biomarker values from the 110 training samples. **c** Random forest model predictions and performances on the prospective validation cohort ($n = 30$) trained on all, not only the significant, restricted biomarker values from the 110 training samples. Biomarker values that fell outside the informative range were replaced with −1 before training and application of the random forest. Our restriction method was used to establish the informative range for every biomarker using only training set samples.

treatment. Clinical responses were assessed using the Response Evaluation Criteria in Solid Tumors (RECIST 1.1)[73]. Patients with progressive disease were categorised as non-responders, whereas those with complete or partial responses, and those with stable disease, were categorised as responders.

**Flow cytometry**

Step-by-step protocols for preparing and analysing clinical samples by flow cytometry can be accessed through Protocol Exchange[74]. Briefly, blood was collected into EDTA-vacutainers by peripheral venepuncture and then delivered to the responsible lab at ambient temperature. Samples were stored at 4 °C for up to 4 h before processing. Whole blood samples were stained using the DURAClone IM T Cell Subsets Tube (Beckman Coulter, B53328). Data were collected using a Navios™ cytometer running Cytometry List Mode Data Acquisition and Analysis Software version 1.3 (Beckman Coulter). An experienced operator performed blinded analyses following a conventional workflow that entailed sample-wise recompensation, arcsinh transformation and rescaling before applying a uniform gating strategy (Supplementary Fig 5).

**Restriction method**

We propose a method for finding biomarkers with high performance in subsets of samples that involves: 1) "restricting" samples into biomarker^HIGH and biomarker^LOW sets for every unique biomarker value; 2) calculating the corresponding restricted receiver operating characteristic (ROC) curve; 3) calculating the area under the restricted ROC curve; 4) adjusting the restricted AUC (rAUC) for sample size; 5) selecting the optimal restriction level, 6) calculating permutation p-values; and 7) reporting performance and significance. This algorithm is implemented as an R package called restrictedROC[36].

To define our nomenclature, we first introduce ROC curve analysis. Let a cutoff $c$ be a real number ($c \in \mathbb{R}$), a continuous biomarker $Y \in \mathbb{R}$ and a grouping of samples into diseased (positive, $D = 1$) and non-diseased (negative, $D = 0$). A sample can be classified as diseased if $Y \geq c$ and non-diseased if $Y < c$. The true positive rate (TPR) and false positive rate (FPR) at cutoff $c$ are defined as $\text{TPR}(c) = P[Y \geq c | D = 1]$ and $\text{FPR}(c) = P[Y \geq c | D = 0]$. The ROC curve relates the TPR and FPR for all possible cutoffs $c$, including $\{\infty, -\infty\}$ (*nb.* compare with Supplementary Fig 16). We can write the value of the ROC curve at any false

positive rate $t$ between 0 and 1 as $\text{ROC}(t) = \text{TPR}(\text{FPR}^{-1}(t))$. For notational simplicity, we introduce $P[Y_D \geq c] := P[Y \geq c | D = 1]$, $P[Y_{\bar{D}} \geq c] := P[Y \geq c | D = 0]$, $S_D(y) := P[Y_D \geq c]$ and $S_{\bar{D}}(y) := P[Y_{\bar{D}} \geq c]$. By substituting TPR and FPR, we get $\text{ROC}(t) = S_D(S_{\bar{D}}^{-1}(t))$. The area under the ROC curve (AUC) is then defined as

$$\text{AUC} = \int_0^1 \text{ROC}(t) dt. \tag{1}$$

Consequently, a perfectly discriminating biomarker with higher values corresponding to the positive class translates to a perfect ROC curve with AUC = 1. An uninformative biomarker has an AUC of 0.5, corresponding to $\text{ROC}(t) = t$ for all values of $t$ between 0 and 1. A perfectly discriminating biomarker but with higher values corresponding to the negative class has an AUC of 0. From a probabilistic point of view, the AUC equals the probability that the biomarker value of a random positive sample will be higher than that of a random negative sample: $\text{AUC} = P[Y_D > Y_{\bar{D}}]$[33,75,76]. The derivation is given in Supplementary Note 3.

Next, we introduce the concept of restricted ROC curves. Our "restriction" is a biomarker value that splits the samples into biomarker[HIGH] and biomarker[LOW] sets. For both sets, we separately calculate "restricted" ROC curves and their corresponding restricted AUC (rAUC). See the supplement for the full derivation. In Supplementary Note 4, we prove that calculating rAUC is identical to scaling a partial AUC (pAUC). Therefore, before we describe our computational method, we consider the (two-way) pAUC[50,77]. The partial AUC (pAUC) is defined as the AUC up to a certain false positive rate $t_0$. Its probabilistic correspondence has been shown[48,76]:

$$\text{pAUC}(t_0) = \int_0^{t_0} \text{ROC}(t) dt = P\left[Y_D > Y_{\bar{D}} | Y_{\bar{D}} > S_{\bar{D}}^{-1}(t_0)\right] \cdot t_0 \tag{2}$$

The pAUC was recently extended to two-way partial AUCs[50]. The two-way partial AUC is defined as the area of the ROC curve between a minimum true positive rate $1 - \alpha$ and a maximum false positive rate $\beta$. This area, shown in Supplementary Fig. 17 as shaded area A, can be written as

$$\text{AUC}_\alpha^\beta = \int_{S_D(S_{\bar{D}}^{-1}(1-\alpha))}^\beta \text{ROC}(t) dt - (1-\alpha)\left(\beta - S_{\bar{D}}\left(S_D^{-1}(1-\alpha)\right)\right) \tag{3}$$

$$= P\left[Y_D > Y_{\bar{D}}, Y_D \leq S_D^{-1}(1-\alpha), Y_{\bar{D}} \geq S_{\bar{D}}^{-1}(\beta)\right] \tag{4}$$

Our restriction method uses two special cases of $\text{AUC}_\alpha^\beta$, shown in Supplementary Fig. 18:

1. The left part of the area under the curve up to a false positive rate $\beta$, which is identical to the pAUC described earlier

$$\text{AUC}_{high}(\beta) = \text{AUC}_{\alpha=1}^\beta = \tag{5}$$

$$= \int_0^\beta \text{ROC}(t) dt \tag{6}$$

$$= P\left[Y_D > Y_{\bar{D}}, Y_{\bar{D}} > S_{\bar{D}}^{-1}(\beta)\right] \tag{7}$$

2. The right part of the area under the curve with at least a true positive rate of $1 - \alpha$

$$\text{AUC}_{low}(\alpha) = \text{AUC}_\alpha^{\beta=1} = \tag{8}$$

$$= \int_{S_D(S_{\bar{D}}^{-1}(1-\alpha))}^1 \text{ROC}(t) dt - (1-\alpha)\left(1 - S_{\bar{D}}\left(S_D^{-1}(1-\alpha)\right)\right) \tag{9}$$

$$= P\left[Y_D > Y_{\bar{D}}, Y_D \leq S_D^{-1}(1-\alpha)\right] \tag{10}$$

Partial AUCs consider only a specific part of the original ROC curve, therefore the interpretation of perfect (AUC = 1) or uninformative (AUC = 0.5) becomes invalid. For pAUC, the following standardisation was proposed to restore this interpretation[51]

$$\text{standardised pAUC} = \frac{1}{2}\left(1 + \frac{\text{pAUC} - \min}{\max - \min}\right) \tag{11}$$

where min is the pAUC given an uninformative biomarker $\left(\min = \frac{\beta^2}{2}\right)$, and max is the pAUC given a perfect biomarker ($\max = \beta$) up to an false positive rate of $\beta$.

In contrast, our restriction method applies the following two scaling factors to any two-way partial $\text{AUC}_\alpha^\beta$

$$\text{rAUC}_\alpha^\beta := \text{AUC}_\alpha^\beta \cdot \frac{1}{\beta - S_{\bar{D}}\left(S_D^{-1}(1-\alpha)\right)} \cdot \frac{1}{S_D\left(S_{\bar{D}}^{-1}(\beta)\right) - (1-\alpha)} \tag{12}$$

Effectively, these two scaling factors rescale the area spanned through $\alpha$ and $\beta$ to 1. Importantly, this is equivalent to calculating rAUC considering only samples with $S_{\bar{D}}^{-1}(\beta) < t < S_{\bar{D}}^{-1}(1-\alpha)$. This has a probabilistic interpretation of

$$\text{rAUC}_\alpha^\beta = P\left[Y_D > Y_{\bar{D}} | S_{\bar{D}}^{-1}(\beta) \leq Y \leq S_{\bar{D}}^{-1}(1-\alpha)\right] \tag{13}$$

Here, the $\text{rAUC}_\alpha^\beta$ is defined in terms of maximum false positive rate $1 - \alpha$ and minimum true positive rate $\beta$. Alternatively, we introduce a "restriction" $r \in \mathbb{R}$ which splits the data into biomarker[HIGH] and biomarker[LOW] sets where $\alpha := 1 - S_D(r)$ and $\beta := S_{\bar{D}}(r)$. With this, our two special cases become

$$\text{rAUC}_{high}(r) = \text{rAUC}_{\alpha=1}^{\beta=S_{\bar{D}}(r)} = \text{AUC}_{high}\left(S_{\bar{D}}(r)\right) \cdot \frac{1}{S_{\bar{D}}(r)} \cdot \frac{1}{S_D(r)} \tag{14}$$

$$\text{rAUC}_{low}(r) = \text{rAUC}_{\alpha=1-S_D(r)}^{\beta=1} = \text{AUC}_{low}\left(1 - S_D(r)\right) \cdot \frac{1}{1 - S_{\bar{D}}(r)} \cdot \frac{1}{1 - S_D(r)} \tag{15}$$

This is equivalent to keeping biomarker[HIGH] samples with values $> r$ ($\text{rAUC}_{high}$) or to keeping biomarker[LOW] samples with values $\leq r$ ($\text{rAUC}_{low}$), then calculating AUC on the restricted dataset. Supplementary Movie 3 uses a hypothetical dataset to visualise the rAUC and show the visual equivalence of our scaling factor compared to restricting the dataset.

More extreme restrictions result in fewer samples, so our estimates of $\text{rAUC}(r)$ become increasingly unreliable; therefore, we adjust $\text{rAUC}(r)$ for sample size after restriction. Here, we leverage the equality of the AUC to the Mann-Whitney U test[33] to calculate the restricted standardised AUC ($\text{rzAUC}_X$) for $X$ either biomarker[HIGH] and biomarker[LOW] sets by calculating the test statistic

$$\text{rzAUC}_X(r) = \frac{\text{rAUC}_X(r) - 0.5}{\sqrt{\text{var}_{H_0}(\text{rAUC}_X(r))}} \tag{16}$$

where $\text{var}_{H_0}(\text{rAUC}_X(r))$ is the variance under the null hypothesis $H_0$ that positive and negative samples are independent and identically distributed. This demands no assumption of normality. Then $\text{var}_{H_0}$ is

given by the following approximation[78,79]:

$$\text{var}_{H_0}\left(\text{rAUC}_X(r)\right) = \frac{m+n+1}{12mn} \qquad (17)$$

where $m$ is the number of positive samples and $n$ is the number of negative samples with biomarker values higher $\left(\text{rzAUC}_{high}\right)$ or lower or equal $\left(\text{rzAUC}_{low}\right)$ than the restriction $r$. With this adjustment, a higher number of samples reduces variance, hence $\text{rzAUC}_x$ becomes more reliable. For a visual example, see Supplementary Fig. 19 where the rAUC and rzAUC are shown for all possible restrictions in terms of the false positive rate. The $\text{rzAUC}_X$ can be negative if the corresponding $\text{rAUC}_X$ is below 0.5, decreases with fewer samples and increases in absolute value the further $\text{rAUC}_X$ is from 0.5. Note that the variance in Eq. (17) is an approximation. Despite this, it provides a reasonable level of accuracy for as few as 6 samples per group. It is computationally infeasible to calculate a Mann-Whitney U test for every possible data-split, therefore we use this approximation for all number of samples.

After calculating the $rzAUC$, we next identify the optimal restriction, which is defined as the highest absolute value of $\text{rzAUC}_{high}$ or $\text{rzAUC}_{low}$. Including more samples would result in a smaller $\text{rAUC}_X$ and therefore smaller $\text{rzAUC}_X$. Excluding more samples would result in an equal or higher $\text{rzAUC}_X$ but also a higher variance and therefore a smaller $\text{rzAUC}_X$. With this restriction, we include some and potentially, but not necessarily, exclude other samples in the calculation of the rAUC. We describe the excluded samples as "unclassifiable" and remove them from the further calculation of usual performance measures like accuracy, specificity, or sensitivity.

Finally, we calculate permutation p-values for the unrestricted AUC and rzAUC. After obtaining the unrestricted AUC for an unrestricted dataset or the $\text{rzAUC}_X$ for an optimised subset of samples, we need to assign a p-value using permutation tests. This is a non-parametric way to determine statistical significance based upon a null hypothesis that class labels assigned to samples are exchangeable[38]. Following this approach, we first calculate unrestricted AUC, $\text{rzAUC}_{high}$ and $\text{rzAUC}_{low}$ using the correct labels. Then we permutate the labels 10,000 times before recalculating unrestricted AUC, $\text{rzAUC}_{high}$ and $\text{rzAUC}_{low}$. To calculate permutation p-values, we use the statmod R package which incorporates a slightly more powerful method than just correcting by $(n_{\text{above}}+1)/(n_{\text{total}}+1)$[80]. $n_{total}$ is the total number of permutations. For the unrestricted permutation p-value, $n_{above}$ is the number of times the permuted unrestricted AUC is above the original unrestricted AUC. Likewise, for the restricted permutation p-value, $n_{above}$ is the number of times either $\text{rzAUC}_{high}$ or $\text{rzAUC}_{low}$ is absolutely higher than the optimal $\text{rzAUC}_X$.

## Multivariate restriction analysis

Our restriction method identifies only a part of the samples as classifiable and cannot make predictions for the unclassifiables. This potentially excludes many samples, so constrains predictive power. To circumvent this problem, we replace the biomarker values of unclassifiable samples with a distinct value (-1) and then apply a random forest. With this substitution, we can predict all given samples, regardless if they are unclassifiable by some biomarkers. In our melanoma dataset, we first downsampled 10,000 CD3$^+$ T cells per sample. We then restricted our set of biomarkers to 84 gates where at least 10% of 110 training samples contained more than 10 counts. Then we calculated the relative proportion of gate cells with respect to either CD4$^+$ CD8$^-$ or CD4$^-$ CD8$^+$ T cells. We also used CD4$^+$CD8$^+$ (double positive), CD4$^-$ CD8$^-$ (double negative), CD4$^+$ CD8$^-$ and CD4$^-$ CD8$^+$ T cell counts, which were expressed as a proportion of the fixed parent gate of 10,000 CD3$^+$ T cells.

For our unrestricted, classical multivariate approach, we used the proportions and counts of all 110 previously published training samples. We then trained a random forest[42] model using the H2O R library[81]

with 1000 trees and 100 bins, a random manual seed for reproducibility of the results of the remaining default parameters. Explicitly, a maximum depth of 20, a minimum number of samples in a node of 1, logloss stopping metric, the number of randomly sampled candidate biomarkers as floor of the square root of 84 (9), a sample rate of 0.632, minimum split improvement of $10^{-5}$ and an automatic histogram type. Finally, we applied the random forest on a prospective cohort of $n = 30$ patients.

For our restricted multivariate approach, we performed a biomarker-wise restriction to samples, and then replaced all unclassifiable biomarker values with -1. We chose this value because all classifiable values are strictly positive as they represent either proportions of CD4$^+$ or CD8$^+$ T cells, or absolute T cell counts. This substitution forces each tree in the random forest to select discriminatory cutoffs within the range of informative biomarker values. We then trained a random forest model with the same settings as for the unrestricted multivariate approach. We finally applied the restriction values obtained from the training set to the prospective validation set, replaced the unclassifiable biomarker values with -1 and applied the random forest to the prospective cohort.

## Synthesising realistic flow cytometry data

Our method to synthesise realistic flow cytometry data is accessible as python[82] package NBNode (v1.1.0) via GitHub[83]. The process of hierarchically gating cells and simulating data with any given effect in any cell population involves five steps. In the following, **bold** letters denote vectors, *italic* letters scalar values and roman multi-letter scalars or functions.

In essence, our approach leverages a Dirichlet process Gaussian mixture model for characterising pre-identified cell populations. Established model-based clustering methodologies such as BayesFlow[54], HDPGMM[53], or NPFlow[55] discern individual cell clusters along with their parameters and weights. In contrast, we only estimate cluster parameters and weights using pre-identified cell populations. Moreover, the hierarchical aspect typically arises from a hierarchy of latent variables rather than from aggregating cell populations according to a predefined gating hierarchy.

In the first step, we applied a uniform manual gating to 48 human peripheral blood samples stained with the DURAClone IM T Cell Subsets Tube (Beckman Coulter GmbH). Data were preprocessed by manually recompensating the samples, removing TIME, and asinh transforming all cell antigen expressions $x$

$$\text{asinh}_{\text{cofactor}}(x) = \text{asinh}\left(\frac{x}{\text{cofactor}}\right) \qquad (18)$$

with the following cofactors: FS INT: 1, FS TOF: 1, SS INT: 1, CD45RA FITC: 1000, CCR7 PE: 2000, CD28 ECD: 2000, PD1 PC5.5: 800, CD27 PC7: 3000, CD4 APC: 4000, CD8 AF700: 10000, CD3 AA750: 500, CD57 PB: 2000, CD45 KrO: 20. Because the channel-wise median fluorescence intensity (MFI) varied between samples, this alone was not sufficient to apply the same gating to all samples. Therefore, we performed a sample-wise rescaling (Supplementary Fig 20 and Supplementary Movie 4). For every cell antigen $x$, we identified the positive and negative population of all cells and found the corresponding $\text{MFI}_x^+$ and $\text{MFI}_x^-$. Using these, the rescaling min-max standardises all cells per sample,

$$\text{rescale}(x) : = \frac{x - \text{MFI}_x^-}{\text{MFI}_x^+ - \text{MFI}_x^-} \qquad (19)$$

leading to a rescaled $\text{MFI}_{\text{rescale}(x)}^+$ of 1 and a rescaled $\text{MFI}_{\text{rescale}(x)}^-$ of 0.

We then applied a standard gating strategy, which is shown schematically (Supplementary Fig 21a) and explicitly for a real-world sample (Supplementary Fig 5). This hierarchical gating of biaxial

scatter plots is effectively a decision tree with 98 "leaf" gates (Supplementary Fig 21a). Each leaf gate corresponds to a terminal gating node and all supraordinate nodes are "intermediate" gates. Every cell must fall into one, and only one, of the subordinate 98 leaf gates.

In the second step, we model the proportion of cells in each leaf gate after uniformly gating all cells from all samples. Specifically, we describe the proportion of cells in each gate according to a Dirichlet distribution Dir($\alpha$) (Supplementary Fig 21b,c). The Dirichlet distribution is a suitable choice after its mass is only on non-negative compositions that sum up to one. Following Minka[84], let $p \in (0,1)^K$ be one random vector of proportions such that $\sum_k^K p_k = 1$ for $k \in \{1, \ldots, K\}$ for K cell populations. In our case, all cells of a sample fall into one and only one of the 98 terminal gates. Therefore, the sum of the cell percentages in each terminal gate adds up to 100%. The probability density under the Dirichlet model with a parameter vector $\alpha \in \mathbb{R}_{>0}^K$ is defined as

$$p(p) \sim \text{Dir}(\alpha_1, \ldots, \alpha_K) = \frac{\Gamma(\sum_k \alpha_k)}{\prod_k \Gamma(\alpha_k)} \prod_k p_k^{a_k - 1} \quad (20)$$

$$\text{with } p_k > 0 \text{ and } \sum_k^K p_k = 1$$

More intuitively, the $\alpha$ parameters can be split into mean proportions per cell population and a precision:

$$m = E[p] = \frac{\alpha}{\sum_k^K \alpha_k} \text{ (mean vector)} \quad (21)$$

$$s = \sum_k^K \alpha_k \text{ (precision)} \quad (22)$$

Hence, a useful explanation of the parameters is that the higher the precision, the more localised the probability becomes around the means. $\alpha_x > \alpha_y$ indicates that, on average, the proportion of cell population $x$ is higher than the proportion of cell population $y$. If $0 < \alpha_k < 1$, the distribution is effectively pushed away from the corresponding cell population. See Supplementary Fig 22 and Supplementary Table 2 for examples of the Dirichlet distribution with K = 3 and different parametrisations of $\alpha$. Plots were created using the R-package dirichlet[85]. We calculate the maximum likelihood of the distribution parameters $\alpha$ with the python dirichlet package[86] based on $N_\text{samples} = 48$ measured cell population proportions $p^{(i)}$ for $i \in \{1, \ldots, N_\text{samples}\}$. In some cell populations and samples there were no cells so the proportion became zero. Because the estimation cannot handle proportions equal to zero, we added a pseudo-proportion to all proportions and normalised to 1 before applying maximum likelihood estimation. With this, the zero-adjusted proportion $p_k^{(i)''}$ of sample $i$ and cell population $k$ becomes

$$p_k^{(i)''} = \frac{p_k^{(i)'}}{\sum_k^K p_k^{(i)'}} \quad (23)$$

$$\text{with } p_k^{(i)'} = p_k^{(i)} + 0.001 \cdot \min(\textit{all proportions}) \quad (24)$$

We end up with a Dirichlet distribution with estimates for the parameter $\hat{\alpha}$

$$\text{Dir}(\hat{\alpha}_1, \hat{\alpha}_2, \ldots \hat{\alpha}_K) \quad (25)$$

In the third step, we build a gating hierarchy using the estimated $\alpha$ parameters corresponding to the leaf nodes. We used the estimated

Dirichlet parameters and manual gating structure to create a probabilistic representation of the gating hierarchy. In this structure, all cells fall into one and only one gate. To calculate intermediate nodes, we sum the estimated $\alpha$ parameters according to the manual gating tree, starting from the bottom and working to the top. Given a Dirichlet distributed variable with K cell populations

$$p(p) = (p_1, \ldots, p_K) \sim \text{Dir}(\alpha_1, \alpha_2, \ldots, \alpha_K) \quad (26)$$

the sum of any two cell populations is again Dirichlet distributed

$$(p_1, \ldots, p_i + p_j, \ldots, p_K) \sim \text{Dir}(\alpha_1, \ldots, \alpha_i + \alpha_j, \ldots, \alpha_K) \quad (27)$$

Therefore, every intermediate or leaf node is described by a Dirichlet distribution. Intuitively, all cells of any gate must fall in one of the subsequent gates and can, therefore, reflect a Dirichlet distribution. To visualise proportions corresponding to these parameters, the decision tree was shaded in red, such that deeper red indicates a higher proportion of cells in that gate (Supplementary Fig 21a).

In the fourth step, we fit a cell antigen distribution using cells from all samples per leaf gate (Supplementary Fig 21d,e). The Dirichlet distributions only describe the number of cells in every gate – that is, a vector of K cell population proportions $p(0,1)^K$. However, a flow cytometry measurement results in a $\mathbb{R}^{n \times m}$ matrix with $n$ cells (rows) and $m$ cell antigens (columns) where every cell comes from a specific cell population. Each such cell population is defined by the $m$ continuous cell antigen expressions. Accordingly, we model the cells for each leaf node $l$ by a multivariate normal distribution $\mathcal{N}(\mu_l, \Sigma_l)$ with mean $\mu_l \in \mathbb{R}^m$ and covariance matrix $\Sigma_l \in \mathbb{R}^{m \times m}$. In the illustrated example, we show the parameters of one gate's normal distribution with the centres of the ellipsoids $\mu_l$ and the shaded areas $\mu_l \pm \sigma$ (Supplementary Fig 21e). We estimated the normal distributions using all cells from $n = 48$ samples. For populations with <2 cells, a covariance matrix was not calculable, so such populations were removed.

In the fifth step, we use the estimated cell population and cell antigen distributions to generate realistic flow cytometry datasets. We use the estimated parameters of the Dirichlet distribution and the normal distributions of each leaf node to generate cells. As shown in Supplementary Fig 23, this simulation involves: (a) drawing a vector $p \in \mathbb{R}^K$ from the estimated Dirichlet distribution $\text{Dir}(\hat{\alpha}_1, \hat{\alpha}_2, \ldots \hat{\alpha}_K)$, which represents the proportion of cells in each leaf node; (b) calculating the number of synthetic cells per leaf node using the expected number of cells for the sample (e.g. 10,000 cells); and (c) Finally, drawing the required number of synthetic cells from the normal distribution of each corresponding leaf node for each sample. By repeating this process for each sample, we generate a synthetic dataset that reflects the underlying population of cells. We visualise our complete decision tree as an interactive online tool (https://vissim.gunthergl.com/) (Supplementary Note 2).

### Imitation of disease-associated effects

We can now introduce any given effect in any given cell population and obtain cells from a realistic synthetic sample. For that, we change the underlying Dirichlet distribution and then sample from the existing normal distributions as before. To change the proportion of cell population $x$, we have to change its parameter $\alpha_x$. However, simply changing $\alpha_x$, e.g. by a factor $f \in \mathbb{R}_{>0}(\alpha_x' := f \cdot \alpha_x)$ also changes the precision. Consequently, the effective change of the population

proportion is different than multiplying with $f$

$$E[p'_x] = \frac{\alpha'_x}{s'} = \frac{f \cdot \alpha_x}{s - \alpha_x + f \cdot \alpha_x} = \frac{f \cdot \alpha_x}{s + (1-f) \cdot \alpha_x} \neq f \frac{\alpha_x}{s} \qquad (28)$$

Therefore, we calculate the new $\alpha_x$ by the share of the expected target proportion in the total old precision and the remaining precision is shared across all other nodes

$$\alpha'_x := \text{target}_\% \cdot s \qquad (29)$$

$$\alpha'_{\text{not } x} := (1 - \text{target}_\%) \cdot s \qquad (30)$$

where "not $x$" corresponds to all nodes which are not the changed node $x$ nor subordinate nodes. Because a single synthetic cell comes from a specific leaf node distribution, we still have to express the changed intermediate node $x$ by its leaf nodes. After parameter $\alpha_k$ of any node is the sum of all leaf node parameters $\alpha_l$ below node $k$, we calculate the new leaf node parameter $\alpha'_l$ as the old $\alpha_l$ multiplied with the ratio of the new and old changed node above

$$\alpha'_l = \alpha_l \frac{\alpha'_x}{\alpha_x} \; or \; \alpha'_l = \alpha_l \frac{\alpha'_{not \, x}}{\alpha_{not \, x}} \qquad (31)$$

This finally leads us to a change in the expected proportion of the target population $x$.

**Reporting summary**
Further information on research design is available in the Nature Portfolio Reporting Summary linked to this article.

## Data availability
The authors declare that all data supporting the findings of this study are available within the paper, its supplementary information files and downloadable files deposited at figshare (https://doi.org/10.6084/m9.figshare.22759076). We created a convenience R-package dataMelanoma for the used data at https://github.com/ggrlab/dataMelanoma[87]. Source data are provided with this paper.

## Code availability
The authors declare that all computer code supporting the findings of this study are available as supplementary information files and downloadable files deposited at figshare (https://doi.org/10.6084/m9.figshare.22759076). The Python package NBNode[83] is accessible at https://github.com/ggrlab/NBNode. The R package restrictedROC[36] is accessible at https://github.com/ggrlab/restrictedROC.

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

## Acknowledgements
The authors thank the Bristol Myers Squibb Foundation for Immuno-Oncology (Award FA-19-009), the Bayerischen Zentrums für Krebsforschung (BZKF; Award BF/04/R/Hutch) and the Bavarian State Ministry for Science and Art's Coronavirus Research Programme for funding this work. We thank Beckman Coulter Life Sciences GmbH for its continuing support of our research. We are grateful to Ian V. Hutchinson (Providence St John's Cancer Institute, CA, USA) and Hansjörg Baurecht (University Hospital Regensburg, Germany) for proof-reading our manuscript. This work was only possible with Erika Ostermeier's outstanding technical support.

## Author contributions
G.G. and J.A.H. designed and performed the project, and wrote the manuscript. K.K. and P.R. performed experiments and analysed data. R.L., V.J.L.M., and R.S. gave expert computational and statistical advice. M.K. gave expert advice about flow cytometry. H.J.S. and E.K.G. gave critical feedback. S.H. provided clinical samples and information, and gave expert Dermatological Oncology opinion.

## Funding

## Competing interests
M.K. is an employee of Beckman Coulter Life Sciences GmbH, a company that manufactures laboratory instruments and reagents. All other authors declare no competing interests.
