## [Peer Review File · Nature Communications]

REVIEWER COMMENTS

Reviewer #1 (Remarks to the Author):

This is an interesting and very well written manuscript describing a new methodology to optimally classify flow cytometry samples (also applicable to other types of datasets). The study clearly articulates and addresses what is a very relevant challenge in immunology, which is the difficulty of classifying individuals based on immunological features due to noise and the unequal variance of immunological features between diseased and healthy individuals. A thorough statistical review is beyond my expertise. I note however, that the bulk of the results correspond to synthetic datasets, with only 1 real-life flow cytometry dataset being analysed. In order to be convinced about the robustness of the methodology, I believe additional real datasets (e.g. publicly available flow cytometry or transcriptional datasets) should be analysed.

Reviewer #3 (Remarks to the Author):

Summary

It is often impossible to find biomarkers that perfectly describe and subsequently predict a phenomenon. Most biomarkers are applicable only to a specific feature range (i.e. very low vs very high, rendering the middle of the range useless for prediction) or, conversely, only a subset of samples are classifiable using a specific biomarker. In this paper the authors present a method for observation restriction, as means to discover biomarkers that are useful for subsets of samples. Many biomarkers might be truly multivariate, in which cases this method is a fantastic addition to the biomarker discovery and classification toolbox - particularly for small data sets where unsupervised learning is not applicable. The paper is extremely well written with the target audience in mind. It was a pleasure to read! I really like the gif versions of the figures and the interactive website as supporting materials. Well done!

Major comments

Given the examples in this work, I am not completely convinced (although I am very close!) that this model is a good way to defining less-than-perfectly discriminatory biomarkers, or if it is a very sophisticated way to overfit a model to spurious feature differences. In order to be convinced, I would like to see the following:

- a) Markers discovered by others shown useful on one or more independent data sets. I think it would be extremely convincing if you improve classification with markers discovered independently by others.
- b) (Multivariate) markers discovered using this approach on a data set, and then validated on one or more independent data sets. You do a nice “fully independent, prospective validation set of 30 patients” for your hepatitis model, but I cannot find the full information about this. Was the training and test data both generated and processed by you? Forgive me if I missed this information somewhere.

Biomarkers can be any measurable feature consistently observed in conjunction with a phenomenon. As such, they technically do not need to have anything to do with the etiology of a disease. If you want to make statements about whether your method discovers etiologically relevant biomarkers, it would be very interesting to see what happens when you shuffle the class labels and rerun the model a number of times. My guess is that you could find sets of discriminatory markers for any random partitions using your model. How many cases out of, say, 1000, do you get an equally good discrimination between random classes? Being able to discriminate random partitions of samples would not in itself disqualify the model from being useful for biomarker discovery, but it would help readers get an idea of whether you discover etiologically informative biomarkers or not.

Minor comments

I definitely appreciate the definitions of “markers” in the beginning of the document, but the term has such a deep history of being used to describe a protein expressed on a specific cell subset, that I found myself confused multiple times when reading the manuscript. I am not adamant about how you define your terms, but just be aware that you risk confusing readers.

One thing that I have discovered many times to affect the usability of population frequencies as biomarkers is the composition of the populations in the samples (i.e. if one population increases, the sum of the rest decrease). Have you looked into this? I think it is at the very least worthy of a discussion point.

You picked a fairly tricky case when you chose to work with flow cytometry, as the definition of features is quite subjective. Whether defined by manual gating or unsupervised methods, choice of Ab clones, protein markers, instruments, normalizations, transformations, and algorithms for analysis may affect results and sometimes interpretations greatly. In my work I have found that to be a very limiting property of flow data in the hunt for robust, generalizable biomarkers. Have you reflected on this? You use a *very* idealized version of manual gating for your proof of concept analyses. I don't think they are particularly comparable to real world gating.

Reviewer #4 (Remarks to the Author):

Restricting datasets to classifiable samples augments discovery of immune disease markers

Glehr and co-author introduce a proposed strategy for discovering disease markers by so-called dataset restriction. They first split their dataset into marker high and marker low, and hence separate their dataset into subsets which are more amenable to classification. I find the article generally well written and engaging to read even though the style is somewhat didactic due to the revisiting of elementary statistics. The article feels very specialized and I found it difficult to see broad interest in the methodology. Though I feel there is a path for this paper to remedy these issues and so I remain positive about potential publication. There is a concern about double-dipping that must be addressed.

After re-reading the manuscript several times, I still find the method and motivation hard to grasp. Does the ROC curve not already tell you which thresholds are discriminative at a particular threshold? Is the issue a reliance on using significance of coefficients in a classification model to decide whether something is a marker? Something can still be useful with markers without a p -value < 0.05 . I believe the issue is that the exposition starts with explaining ROC curves rather than explaining the end goal of the analysis and how ROC analysis plays a part in this.

Broad Interest:

The case-study and examples are limited to a very narrow domain application. The authors mention that their method can be applied more broadly. They should remonstrate this level of applicability within the manuscript. If the applicability of the method is as wide as they claim, then this should not be difficult.

Is the unclassifiable region generalisable outside the dataset of interest. For example, would the same threshold be used each time or is it dependent on the cohort. If the latter it has to see how the method would be applied in practice.

Technical issues:

The authors heavily rely on Gaussian distribution for demonstration but from what I understood many of the values they observed can only be positive and so Gaussian would be an odd assumption here. Furthermore, if the method is applicable more widely to say transcriptomics, count distributions would need to be considered.

The use of the Dirichlet distribution in the simulation is welcome but it has limitations in that the proportions are uncorrelated, which is unlikely in practice. I would suggest the authors consider more elaborate simulation scenarios such as with the logit normal.

I am concerned about double-dipping as the authors appear to use the data twice which will cause p -values and estimates of predictive performance to be uncalibrated. I may be wrong in the way the method is explained but the method first defines an unclassifiable region via prediction (the first dip) and then proceeds to apply classification methods and multivariate statistical methods (the second dip). They may already be handling this issue but it is not exposed carefully so I cannot fully understand. The most convincing way would be to plot estimated FDR against actual FDR for several examples. Including where the model is mis-specified. If the authors are not familiar the issue is sometimes called selective inference.

Some of the permutation p -values are equal to 0, which should never happen. The authors may wish to revisit these computations and explain how they are calculated.

Stylist comments:

The style is somewhat patronizing in places. For example, comments such as: "To do this, we must first disambiguate some key terms with different meanings for immunologists and computer scientists. " are not necessary - simply define the concepts you think will be confusing.

On the opposite side, the use of symbols such as \forall \in etc are not appropriate for manuscripts and better for the blackboard. This is discouraged.

The discussion is long and repetitive and could be shortened.

Response to reviewers' comments

We are grateful to the three reviewers for their challenging comments and suggestions for improving our manuscript. All three reviewers requested that we apply our restriction method to external datasets to confirm its utility and applicability to other data types. Accordingly, we now present results from the following repurposed datasets:

Type of data	Reference to data source	Implementation
Proteomic	Harel et al. 2019. Cell 179(1), 236-250.	Fig.S9&10
Microbiomic	Lee et al. 2022. Nature Med. 28(3), 535-544.	Fig.S11
Mass cytometry	Lonzano et al. 2022. Nature Med. 28(2), 353-362.	Fig.S12
Transcriptomic	Zhang et al. 2022. Genome Medicine. 14(1), 45.	Fig.S13&14

New univariate markers were discovered in each case. We describe the new results on I.391-394 and I.417-438.

We were concerned by comments from reviewers 3 and 4 about nomenclature. We have now substituted “marker” with “biomarker” to mean a sample-related property with predictive relevance. We now use “cell antigen” to mean a cell-associated property measured by flow cytometry.

Some typographical errors were corrected during revision. All changes from the original submission are marked.

Reply to Reviewer 1

Reviewer #1 (Remarks to the Author):

This is an interesting and very well written manuscript describing a new methodology to optimally classify flow cytometry samples (also applicable to other types of datasets). The study clearly articulates and addresses what is a very relevant challenge in immunology, which is the difficulty of classifying individuals based on immunological features due to noise and the unequal variance of immunological features between diseased and healthy individuals. A thorough statistical review is beyond my expertise. I note however, that the bulk of the results correspond to synthetic datasets, with only 1 real-life flow cytometry dataset being analysed. In order to be convinced about the robustness of the methodology, I believe additional real datasets (e.g. publicly available flow cytometry or transcriptional datasets) should be analysed.

Thank you for your positive appraisal. Your comments about demonstrating the utility of our restriction method using external datasets are completely valid. As described above, we've now applied our restriction method to repurposed datasets published by other groups, including proteomic, mass cytometric, microbiomic and transcriptomic data. New univariate markers were discovered in each case. Furthermore, we were able to improve upon the multivariate predictive models built by Zhang et al. from a meta-dataset compiled from 10 independent transcriptomic studies. This implies that non-informative ranges, bounded by the restriction value, might be generalizable properties of biomarkers.

As with any biomarker discovery approach, the robustness of results generated using our restriction method must be confirmed by validation studies. However, in our revised manuscript, we now show that restriction typically (but not always) leads to superior models (Fig.S15).

Reply to Reviewer 3

Reviewer #1 (Remarks to the Author):

Summary: It is often impossible to find biomarkers that perfectly describe and subsequently predict a phenomenon. Most biomarkers are applicable only to a specific feature range (i.e. very low vs very high, rendering the middle of the range useless for prediction) or, conversely, only a subset of samples are classifiable using a specific biomarker. In this paper the authors present a method for observation restriction, as means to discover biomarkers that are useful for subsets of samples. Many biomarkers might be truly multivariate, in which cases this method is a fantastic addition to the biomarker discovery and classification toolbox - particularly for small data sets where unsupervised learning is not applicable. The paper is extremely well written with the target audience in mind. It was a pleasure to read! I really like the gif versions of the figures and the interactive website as supporting materials. Well done!

Thank you for your positive comments.

Major comments:

Given the examples in this work, I am not completely convinced (although I am very close!) that this model is a good way to defining less-than-perfectly discriminatory biomarkers, or if it is a very sophisticated way to overfit a model to spurious feature differences. In order to be convinced, I would like to see the following:

a) Markers discovered by others shown useful on one or more independent data sets. I think it would be extremely convincing if you improve classification with markers discovered independently by others!

As detailed above, we now applied our restriction method to repurposed datasets published by other groups, including proteomic, mass cytometric, microbiomic and transcriptomic data (Fig.S9-13). New univariate markers were discovered in each case.

b) (Multivariate) markers discovered using this approach on a data set, and then validated on one or more independent data sets. You do a nice “fully independent, prospective validation set of 30 patients” for your hepatitis model, but I cannot find the full information about this. Was the training and test data both generated and processed by you? Forgive me if I missed this information somewhere.

We apologize for this ambiguity. Here, the prospective validation set was data generated in our laboratory. It is independent in the sense that training set samples were collected

between OCT-2016 and JUN-2021, whereas validation set samples were collected between JUN-2021 and JAN-2023. We accept that our validation set isn't "fully" independent – please see changes on I.402-403.

Biomarkers can be any measurable feature consistently observed in conjunction with a phenomenon. As such, they technically do not need to have anything to do with the etiology of a disease. If you want to make statements about whether your method discovers etiologically relevant biomarkers, it would be very interesting to see what happens when you shuffle the class labels and rerun the model a number of time. My guess is that you could find sets of discriminatory markers for any random partitions using your model. How many cases out of, say, 1000, do you get an equally good discrimination between random classes? Being able to discriminate random partitions of samples would not in itself disqualify the model from being useful for biomarker discovery, but it would help readers get an idea of whether you discover etiologically informative biomarkers or not.

If we interpret this question correctly, you are interested in p-value distributions of univariate tests. In our univariate analyses, we rely upon permutation p-values corrected for multiple testing by the false discovery rate to identify significant features. Therefore, we expect some features to be false positive results. As you requested, we performed label permutations to test the idea that our restriction method would somehow select or create discriminatory features for random sample partitions – please refer to Supplementary Figure 7. Here, we investigated the CD4⁺ Tem population with hepatitis labels and permuted the labels 10k times. Setting a p-value significance level of 0.1 leads to 1033 significant permutations for the global AUC and 987 significant permutations for the restricted AUC. In Fig.6, we show uncorrected p-values, but everywhere else we report multiple-testing corrected p-values. In the given setting of 10k label permutations shown in Fig.S7, no p-value remained significant after correction for multiple testing with the false discovery rate.

Minor comments

I definitely appreciate the definitions of "markers" in the beginning of the document, but the term has such a deep history of being used to describe a protein expressed on a specific cell subset, that I found myself confused multiple times when reading the manuscript. I am not adamant about how you define your terms, but just be aware that you risk confusing readers.

Following your recommendation, we have substituted "marker" with "biomarker" to mean a sample-related property with predictive relevance. We now use "cell antigen" to mean a cell-associated property measured by flow cytometry. Please see highlighted changes throughout the manuscript.

One thing that I have discovered many times to affect the usability of population frequencies as biomarkers is the composition of the populations in the samples (i.e. if one population increases, the sum of the rest decrease). Have you looked into this? I think it is at the very least worthy of a discussion point.

We strongly agree that choice of denominator populations greatly affects the informativeness and stability of cell subset frequencies as biomarkers. However, it is unclear to us whether this affects the informative range of biomarkers or not, hence the outcome of restriction. We decline to speculate too much in the Discussion.

You picked a fairly tricky case when you chose to work with flow cytometry, as the definition of features is quite subjective. Whether defined by manual gating or unsupervised methods, choice of Ab clones, protein markers, instruments, normalizations, transformations, and algorithms for analysis may affect results and sometimes interpretations greatly. In my work I have found that to be a very limiting property of flow data in the hunt for robust, generalizable biomarkers. Have you reflected on this? You use a *very* **idealized** version of manual gating for your proof of concept analyses. I don't think they are particularly comparable to real world gating.

Conventional flow cytometry from peripheral blood samples is a fast, relatively inexpensive and accessible technology. Compared to other single cell technologies, we believe it has great untapped potential as a clinical-decision making tool. The comparability of data from separate instruments remains a major obstacle, despite advances in technical standardization, calibration and data normalization. We hope you agree that discussion of these key issues is beyond the scope of our current manuscript.

We completely accept your point about our idealized gating strategy and recognize its obvious limitations in real-world diagnostic applications. The advantage of a hard-rules gating in the present context is to eliminate operator-dependent errors.

Reply to Reviewer 4

Reviewer #4 (Remarks to the Author):

Glehr and co-author introduce a proposed strategy for discovering disease markers by so-called dataset restriction. They first split their dataset into marker high and marker low, and hence separate their dataset into subsets which are more amenable to classification. I find the article generally well written and engaging to read even though the style is somewhat didactic due to the revisiting of elementary statistics. The article feels very specialized and I found it difficult to see broad interest in the methodology. Though I feel there is a path for this paper to remedy these issues and so I remain positive about potential publication. There is a concern about double-dipping that must be addressed.

Our manuscript identifies and proposes a solution to a commonly ignored problem in biological research using “omics” datasets with a large number of features – namely, that features of known mechanistic relevance are often not identified by conventional statistical approaches. This problem partly stems from the behaviour of biomarkers in disease, which alters the dispersion of feature expression between healthy and diseased patient populations, leading to informative and non-informative ranges. Our restriction method identifies the largest possible range of informative biomarker values using an iterative approach to define the optimal standardized restricted AUROC. Admittedly, this is a heuristic solution. Unfortunately, we cannot presently explain what kind of variability causes non-informativeness or directly calculate the restriction value.

This manuscript is important because it identifies non-informative ranges of biological features as a major limitation in modern immunological research. The practical applications of our method are important because it potentially increases the “findability” of features in clinical datasets, which are often limited in size by technical costs or scarcity of patient samples. We wrote this paper with an audience of immunologists in mind, which means that we must explain some statistical concepts. The manuscript was proof-read by immunologists (PR, KK, MK, HJS, EKG, SH) and computational scientists (GG, RL, VLM, RS) to find the best compromise in content.

After re-reading the manuscript several times, I still find the method and motivation hard to grasp. Does the ROC curve not already tell you which thresholds are discriminative at a particular threshold?

Yes, the ROC curve relates the TPR and FPR for every value in the dataset, so indeed tells you about the discriminatory performance of a biomarker at a particular threshold. However,

the restriction value is completely different from the discriminatory value. Our restriction method identifies the largest informative range of a biomarker. The restriction value is the boundary between the informative and non-informative measurable ranges.

We argue that the optimal discriminatory value of a biomarker should always lie within the informative range. When building multivariate models, random variability contributed by samples with biomarker values in the non-informative range can lead to misplacement of discriminatory threshold. We suggest this is why random forest models build with restricted datasets are superior to those built with unrestricted (global) datasets.

[B] Is the issue a reliance on using significance of coefficients in a classification model to decide whether something is a marker?

No, the goal of our restriction method is to identify the informative range for a marker, which then determines whether any given sample is classifiable or unclassifiable. Using only information from classifiable samples (i.e. those with biomarker values within the informative range) to assess discriminative performance identifies better performing univariate and multivariate predictors.

[C] Something can still be useful with markers without a p-value < 0.05 .

Yes, we agree that markers with any univariate p-value can be useful in building multivariate models. The purpose of our restriction method as a pre-processing step in multivariate modelling is to use information only from samples with marker values within the informative range, or otherwise set the marker value to a constant. This removes signal variability from non-informative samples. As we demonstrate, this improved the performance of our models on unseen samples in two example cases. Please note, we use all features in our multivariate models, which we now emphasize in the legend of Fig. 8.

[D] I believe the issue is that the exposition starts with explaining ROC curves rather than explaining the end goal of the analysis and how ROC analysis plays a part in this.

Although we fully understand the reviewer's perspective, the starting point of our work was recognizing that disease-associated markers often give rise to skewed ROC curves, reflecting the presence of informative and non-informative ranges. To communicate this point to immunologists and clinicians, we must introduce the basic concept of ROC curves because they aren't commonly understood in the field.

The case-study and examples are limited to a very narrow domain application. The authors mention that their method can be applied more broadly. They should remonstrate this level of

applicability within the manuscript. If the applicability of the method is as wide as they claim, then this should not be difficult.

As we've said, this is an immunology paper that identifies a general problem from a concrete example. Nevertheless, we agree with the reviewer that extending our method to other data acquired with different technologies increases its impact. We now provide examples of our restriction method applied to repurposed datasets. Specifically, proteomic data (Fig S9&10), microbiomic data (Fig S11), mass cytometry data (Fig S12) and transcriptomic data (Fig S13&14).

Is the unclassifiable region generalizable outside the dataset of interest. For example, would the same threshold be used each time or is it dependent on the cohort. If the latter it has to see how the method would be applied in practice.

This is an extremely interesting question. We have no precise mathematical definition of the restriction point. Furthermore, we don't fully understand how disease-related variability in biomarker expression actually relates to non-informative ranges. Consequently, from a theoretical perspective, it's hard to say how difference in the composition of patient populations or technical considerations might affect the generalizability of restriction values. On the other hand, we see no reason why the restriction point shouldn't be generalizable to the same extent that discriminatory thresholds can be applied across different datasets.

Please refer to the new results presented in Supplementary Figure 13-15. Here, transcriptomic data were repurposed from 10 independent studies. Generally, we find an improved multivariate model performance after applying restriction. Because a restriction value established with only training data improved the performance in the identified subset of samples and the performance of multivariate models in the independent validation and test set, we infer that restriction values are generalizable.

In Supplementary Fig 13, we established the restriction values on the predefined training set of 618 samples. Using this restriction value on the validation and test set improved the AUC on the restricted part.

In Supplementary Fig 14, we used the predefined training/validation/test splits and built random forest models with or without restriction on only the training set. We then report the AUCs with (blue) and without (red) restriction on the validation and test sets with and without random forest hyperparameter optimization.

In Supplementary Fig 15, we repeatedly split the total 921 samples into training (70%) and test (30%) sets. We then built random forest models with and without restriction

preprocessing, generally finding an improved performance of the AUC on the completely left out test set.

The authors heavily rely on Gaussian distribution for demonstration but from what I understood many of the values they observed can only be positive and so Gaussian would be an odd assumption here. Furthermore, if the method is applicable more widely to say transcriptomics, count distributions would need to be considered.

Another challenging question! Our values from flow cytometry, being proportions of cell populations in a sample, can indeed only be positive. However, as ROC curves can be calculated from any two distributions, we worked with Gaussian distributions throughout the manuscript as an easy and familiar example. We maintain that it's fair to use Gaussian distributions to understand if restriction performs as expected and to define when it actually improves predictions.

Still, we agree it's important to extend our examples to other distributions. Taking count data as an additional example, we've now applied restriction to four differently parametrized negative binomial distributions – please refer to Supplementary Figure 3. In the simplified case of an equal dispersion parameter across conditions (as implemented in DEseq2 or EdgeR) the informative range of markers is limited as in the Gaussian example of a subset of samples with different marker values (Fig.S8A-C, Fig 5). However, when the dispersion parameter is different between the two compared populations, we observe a clearly limited informative range (Fig.S8D). Following the impact of unequal dispersion parameters in contrast to only differences in variances of two distributions, we replaced unequal “variance” with “variability” or “dispersion” throughout the manuscript.

The use of the Dirichlet distribution in the simulation is welcome but it has limitations in that the proportions are uncorrelated, which is unlikely in practice. I would suggest the authors consider more elaborate simulation scenarios such as with the logit normal.

The reviewer correctly identifies a limitation of using the Dirichlet distribution. The frequencies of many leucocyte subsets are likely to show interdependencies that are not properly reflected in our simulation. For instance, chronic immune activation leading to accumulation of CD4⁺ T_{EMRA} might be associated with accumulation of CD8⁺ T_{EMRA} too, which wouldn't be described by our model. The reviewer makes an excellent suggestion that using the logit-normal distribution might allow more biologically realistic simulation of flow cytometry data. However, this approach has significant downsides in our particular application.

Our simulation relies upon a manual gating tree as a foundational framework. We employ a Dirichlet distribution where each of the k leaf nodes is represented by a concentration parameter. Because summing two Dirichlet distributions results in a Dirichlet distribution, we are able to express intermediate nodes as Dirichlet distributions. At all levels of the simulated gating tree, the joint concentration parameter of intermediate nodes is determined by the sum of concentration parameters from all child leaf nodes. In contrast, when dealing with the logit-normal distribution, there is no simple way to aggregate leaf nodes into intermediate nodes. This is a major theoretical roadblock.

One possible way around this obstacle is estimating a new logit-normal distribution for every possible combination of nodes. Unfortunately, this disrupts the cell simulation because attempting to model only a single normal distribution for each merged intermediate node seriously skews the characteristics of the simulated cells. Apart from this issue, the sheer number possible node combinations presents a formidable computational problem.

In our Dirichlet-based approach, we simulate cells from the leaf nodes exclusively. Each leaf node is described by a multivariate normal distribution that characterizes cell features. It's important to note that the Dirichlet distribution governs solely the proportion of cells originating from each leaf node. When we adjust the concentration parameter of an intermediate node, the effect is confined to altering the concentration parameters of the leaf nodes only. Consequently, any adjustment only affects the proportion of cells associated with each node, preserving the original location parameters (multivariate normal distribution) of cells. This preservation is a vital advantage in ensuring the accurate simulation of cells from each cellular subpopulation.

Our primary motivation for developing our simulation method was to generate more accurate datasets giving rise to skewed ROC curves. In this particular application, our inability to simulate an accurate correlation structure between cell populations does not affect the core findings presented in this paper. Our point is demonstrated in Figure 6A-D, where we focused on alterations to the intermediate cell population of CD4⁺ T_{EM} cells. We note that other groups used Dirichlet distributions for analogous purposes [Johnsson et al., 2016; Lin & Hejblum, 2021].

Please see text changes in the discussion of the manuscript, p25/26

I am concerned about double-dipping as the authors appear to use the data twice which will cause p-values and estimates of predictive performance to be uncalibrated. I may be wrong in the way the method is explained but the method first defines an unclassifiable region via prediction (the first dip) and then proceeds to apply classification methods and multivariate

statistical methods (the second dip). They may already be handling this issue but it is not exposed carefully so I cannot fully understand. The most convincing way would be to plot estimated FDR against actual FDR for several examples. Including where the model is mis-specified. If the authors are not familiar the issue is sometimes called selective inference.

The reviewer is concerned that we used information from the validation set samples to build our predictive models. However, no information from the validation set was used to calculate a restriction value for training data. We first split the dataset into training and validation sets. The restriction value was calculated using only training data. Then, a random forest was created using only the restricted training data. Finally, we applied the restriction value and predictive model from the training set to the validation and test sets. This is a clean approach with no double-dipping. We apologize for the lack of clarity in our original submission, which we've now rectified through changes on l.407-411 and the legend of Fig 8.

Some of the permutation p-values are equal to 0, which should never happen. The authors may wish to revisit these computations and explain how they are calculated.

Thank you for this valuable comment. We've now incorporated the ideas and solution from Phipson and Smyth (2010) who developed a correction for the permutation p-value such that its power is higher than just adding 1 to the number of performed tests and the number of statistic values exceeding the observed statistic. This change was implemented throughout the manuscript, wherever permutation p-values are presented:

- **Code:** Reported as supplementary material and the restrictedROC package
- **Figures:** Fig.5 – simulated values; Fig.6 – all plots; Fig.8 – significance of the gated subsets remained unchanged.
- **Text:** Methods – p35, l747-753. Description of permutation p-values; Results – as highlighted.

Stylist comments:

The style is somewhat patronizing in places. For example, comments such as, "To do this, we must first disambiguate some key terms with different meanings for immunologists and computer scientists," are not necessary - simply define the concepts you think will be confusing. On the opposite side, the use of symbols such as \forall etc are not appropriate for manuscripts and better for the blackboard. This is discouraged. The discussion is long and repetitive and could be shortened.

Writing a paper that's readable for immunologists and computer scientists was difficult. We wrestled with terminology. As Reviewer 3 says, the term "marker" is used by flow cytometrists to mean a molecular property of a cell, whereas most others understand it as a property of clinical samples that predicts disease. We apologize if our style comes across as patronizing, we only mean to be emphatic. We tried to adjust the tone throughout the paper.

With this long paper, we prefer to stick with the inline notation because it makes the manuscript more readable.

REVIEWERS' COMMENTS

Reviewer #1 (Remarks to the Author):

The authors have adequately addressed my main concern, which regarded the lack of independent non-synthetic validation datasets. Altogether, the analytical solution proposed in this manuscript appears to have wide utility for immunology studies and should be of interest to a wide audience.

Reviewer #3 (Remarks to the Author):

Thank you very much for the thorough replies to my comments! I remain highly impressed with this work and the writing style of the manuscript.

In general, I am very happy with the way you addressed the comments. Only minor comments remain:

I wholeheartedly agree with you that the potential of flow cytometry as a diagnostic tool is under-appreciated. I think that so much more can be achieved with this technology owing to its cost efficiency, massive amounts of existing data, and standardized, wide-spread instruments. However, I must maintain that biomarkers discovered with your method (or any other method, for that matter) are only useful insofar as they are transferable across cohorts and data sets. I can accept your idealized proof of concept, and certainly appreciate why you performed your analysis the way you did, but my position is still that these challenges and limitations should at least be mentioned if you wish to make statements about applicability your method outside of a data sets produced under near identical technical conditions.

Reviewer #3 (Remarks on code availability):

I reviewed the original code, but have not explored it after revisions. The formalities appear to be in order.

Reviewer #4 (Remarks to the Author):

The manuscript is much improved and I approve of publication after some minor issues. The authors have clarified the motivation and the addition of the other examples has greatly improved the broad applicability of the method.

Minor Issues

Line 543 starting "In theory". I'm not really sure what the authors intended to state here.

Line 635: Here or elsewhere, it is perhaps worth stating how data restriction would work in the case of a random-effects or if further work is required.

Line 707: I'm not sure your audience will know what \forall or blackboard bold R means, it is no effort to write this out and then the mathematical abbreviation the first time. Again with \forall later.

Line 711: The FPR was never defined

Line 712: I'm confused by the definition of the TPR here - I suspect there are typos. The meaning of \bar{D} is unclear.

Equation (2) is somewhat confusing without explicitly stating that t_0 is the "certain false positive rate"

Line 735: α and β should be defined so that the calculations follow.

Equation (9): I think there is a missing minus sign between the differential and $(1 - \alpha)$. I think (10) is still correct.

Before equation (17). Then var. is given by the following approximation: (At the moment the sentence doesn't make sense and the equality isn't true).

Equation (732) stick with \rightarrow use

Equation (747) interchangeable \rightarrow exchangeable. I think it is good to be precise here.

Equation (837) worth noting that this is equivalent to an empirical Bayesian approach and there are many references you can point to.

Response to reviewers' comments

We are grateful to all three reviewers for greatly improving our manuscript.

Apart from new insertions in response to the editors' and reviewers' comments, we corrected the manuscript for style and formatting. All changes from the original submission are highlighted in the track-changes version. We also provide an unmarked version. Line references below refer to the track-changes version.

Reply to Reviewer 1

Reviewer #1 (Remarks to the Author):

The authors have adequately addressed my main concern, which regarded the lack of independent non-synthetic validation datasets. Altogether, the analytical solution proposed in this manuscript appears to have wide utility for immunology studies and should be of interest to a wide audience.

Thank you for your positive appraisal. Your instruction to apply our method to independent datasets has certainly strengthened our manuscript. We are grateful for your guidance.

Reply to Reviewer 3

Reviewer #3 (Remarks to the Author):

Thank you very much for the thorough replies to my comments! I remain highly impressed with this work and the writing style of the manuscript.

Thank you for your positive comments. Your input was very helpful for improving the paper.

In general, I am very happy with the way you addressed the comments. Only minor comments remain:

I wholeheartedly agree with you that the potential of flow cytometry as a diagnostic tool is under-appreciated. I think that so much more can be achieved with this technology owing to its cost efficiency, massive amounts of existing data, and standardized, wide-spread instruments. However, I must maintain that biomarkers discovered with your method (or any other method, for that matter) are only useful insofar as they are transferable across cohorts and data sets. I can accept your idealized proof of concept, and certainly appreciate why you

performed your analysis the way you did, but my position is still that these challenges and limitations should at least be mentioned if you wish to make statements about applicability of your method outside of a data set produced under near identical technical conditions and materials.

We accept your point. A new paragraph was inserted into our Discussion from L.546 onwards. For your interest, we recently generated a large dataset of paired flow cytometry samples measured on different cytometers. We are repeating our data collection at a fully independent, 3rd-party site. We aim to release this dataset, which includes clinical and technical information, as a public resource for groups interested in benchmarking predictive algorithms by late summer.

Reply to Reviewer 4

Reviewer #4 (Remarks to the Author):

The manuscript is much improved and I approve of publication after some minor issues. The authors have clarified the motivation and the addition of the other examples has greatly improved the broad applicability of the method.

Thank you for again reading our manuscript so closely. We greatly appreciate your efforts in improving our manuscript.

Line 543 starting “In theory”. I'm not really sure what the authors intended to state here.

Thanks. We removed that sentence. L518-520

Line 635: Here or elsewhere, it is perhaps worth stating how data restriction would work in the case of a random-effects or if further work is required.

There was disagreement between the authors about the meaning of this question. If you asked whether random sampling errors could be falsely identified as an informative range of a non-informative biomarker, then yes. Like any other statistical test, dataset restriction will return Type I errors, which we seek to limit using the FDR method. As with any biomarker discovery approach, a separate validation set is the best way to control false-positive results – which is the approach taken in this work.

Alternatively, you might've asked about controlling for unobserved heterogeneity in our datasets using random effects models. Applying a random effects model usually requires

some knowledge about the structure of your data, such as what subgroups exist within a class and which samples belong to which subgroups. In contrast, restriction doesn't require any prior knowledge about subgroups. Therefore, we think that restriction and random effects models are solving different problems. Please see the paragraph inserted into our Discussion on L.618-634

Line 707: I'm not sure your audience will know what \forall or blackboard bold \mathbb{R} means, it is no effort to write this out and then the mathematical abbreviation the first time. Again with \forall later.

Thanks. We added "Let a cut-off c be a real number ($c \in \mathbb{R}$)" and replaced the \forall by "for all values of t between 0 and 1". Please see L.704-705 and elsewhere.

Line 711: The FPR was never defined

We thank the reviewer for these remarks, that was actually a typo. The second "TPR", being $\text{TPR} = P[Y \geq c \mid D=0]$ actually described the false positive rate (FPR). Please see L.707-709

Line 712: I'm confused by the definition of the TPR here - I suspect there are typos. The meaning of \bar{D} is unclear.

We clarified the paragraph with respect to \bar{D} . Our notation followed the idea: $P[Y_{\bar{D}} \geq c] := P[Y \geq c \mid D = 0]$ and $P[Y_D \geq c] := P[Y \geq c \mid D = 1]$. Additionally, we introduced $S_D(y) := P[Y_D \geq c]$ and $S_{\bar{D}}(y) := P[Y_{\bar{D}} \geq c]$, simplifying notation later. L.707

Equation (2) is somewhat confusing without explicitly stating that t_0 is the "certain false positive rate"

We thank the reviewer for this comment. Please see changes on L.733-734

Line 735: α and β should be defined so that the calculations follow.

We thank the reviewer for the comment. We clarified the sentence on L.736

Equation (9): I think there is a missing minus sign between the differential and $(1 - \alpha)$. I think (10) is still correct.

This is indeed true, thanks for pointing that out. Please see L743

Before equation (17). Then var.. Is given by the following approximation: (At the moment the sentence doesn't make sense and the equality isn't true).

Thank you. The sentence was changed as suggested on L.770

Equation (732) stick with -> use

Thank you. The sentence was changed as suggested on L.780

Equation (747) interchangeable -> exchangeable. I think it is good to be precise here.

Thank you. The sentence was changed as suggested on L.795

Equation (837) worth noting that this is equivalent to an empirical Bayesian approach and there are many references you can point to.

We agree that we essentially used an empirical Bayes approach. We added a paragraph on L.845-852:

In essence, our approach leverages a Dirichlet process Gaussian mixture model for characterizing pre-identified cell populations. Established model-based clustering methodologies such as BayesFlow, HDPGMM, or NPFlow discern individual cell clusters along with their parameters and weights. In contrast, we only estimate cluster parameters and weights using pre-identified cell populations. Moreover, the hierarchical aspect typically arises from a hierarchy of latent variables rather than from aggregating cell populations according to a predefined gating hierarchy.